# Deletion of the *Plasmodium falciparum* exported protein PTP7 leads to Maurer's clefts vesiculation, host cell remodeling defects, and loss of surface presentation of EMP1

Olivia M. S. Carmo[1], Gerald J. Shami[1], Dezerae Cox[1], Boyin Liu[1], Adam J. Blanch[1], Snigdha Tiash[1], Leann Tilley[1‡], Matthew W. A. Dixon[2,3‡]*

**1** Department of Biochemistry and Pharmacology, Bio21 Molecular Science and Biotechnology Institute, University of Melbourne, Parkville, Australia, **2** Department of Infectious Diseases, Peter Doherty Institute for Infection and Immunity, University of Melbourne, Melbourne, Australia, **3** Division of Infectious Diseases and Immune Defence, Walter and Eliza Hall Institute of Medical Research, Parkville, Australia

‡ These authors are joint senior authors on this work.
* matthew.dixon@unimelb.edu.au

## Abstract

Presentation of the variant antigen, *Plasmodium falciparum* erythrocyte membrane protein 1 (EMP1), at knob-like protrusions on the surface of infected red blood cells, underpins the parasite's pathogenicity. Here we describe a protein PF3D7_0301700 (PTP7), that functions at the nexus between the intermediate trafficking organelle, the Maurer's cleft, and the infected red blood cell surface. Genetic disruption of PTP7 leads to accumulation of vesicles at the Maurer's clefts, grossly aberrant knob morphology, and failure to deliver EMP1 to the red blood cell surface. We show that an expanded low complexity sequence in the C-terminal region of PTP7, identified only in the *Laverania* clade of *Plasmodium*, is critical for efficient virulence protein trafficking.

## Author summary

We describe a malaria parasite protein, PTP7, involved in virulence factor trafficking that is associated with Maurer's clefts and other trafficking compartments. Upon disruption of the PTP7 locus, the Maurer's clefts become decorated with vesicles; the knobby protrusions on the host red blood cell surface are fewer and distorted; and trafficking of the virulence protein, EMP1, to the host red blood cell surface is ablated. We provide evidence that a region of PTP7 with low sequence complexity plays an important role in virulence protein trafficking from the Maurer's clefts.

## Introduction

*Plasmodium falciparum* causes more than 200 million malaria infections every year, killing more than 600,000 people [1]. Central to the ability of *P. falciparum* to maintain an infection

**Data Availability Statement:** The mass spectrometry proteomics data have been deposited to the ProteomeXchange Consortium via the PRIDE [74] partner repository with the dataset identifier PXD027566. All quantitative and qualitative microscopy data and analysis scripts are available from open-access Zenodo repositories (data DOI: 10.5281/zenodo.5146871; analysis scripts DOI: 10.5281/zenodo.5147885).

**Funding:** LT is a Georgina Sweet, Australian Research Council Laureate Fellow (FL150100106) (http://www.arc.gov.au). MWAD and LT thank the National Health and Medical Research Council (1098992) (https://www.nhmrc.gov.au) for funding this work. The funders had no role in study design, data collection and analysis, decision to publish, or preparation of the manuscript.

**Competing interests:** The authors have declared that no competing interests exist.

and cause disease, is the invasion and remodelling of host red blood cells (RBCs). Maturation of the parasite inside the RBC is accompanied by striking changes in the surface topology of the infected RBC and a marked loss of cellular deformability [2,3]. One key modification is the assembly of ~90 nm diameter structures, called knobs, at the infected RBC surface [4]. The knob structure acts as an elevated platform at the RBC surface for presentation of the major virulence protein, *P. falciparum* erythrocyte membrane protein-1 (EMP1), which mediates binding of infected RBCs to endothelial ligands [5–8].

The trafficking of EMP1 and other virulence determinants beyond the confines of the parasite is mediated by the *Plasmodium* translocon of exported proteins (PTEX) that is present in the parasitophorous vacuole membrane (PVM), with help from a second complex termed the exported protein interacting complex (EPIC) [9–12]. Once exported across the PVM, EMP1 is thought to be trafficked across the RBC cytoplasm as a soluble, chaperoned complex; and inserted into the membrane bilayer of the Maurer's clefts—an intermediate trafficking compartment in the RBC cytoplasm [12–14]. The mechanisms controlling EMP1 insertion into the Maurer's clefts and its subsequent forward trafficking and delivery to the RBC membrane remain unclear. Of interest, coated electron-dense vesicles (EDVs) and uncoated vesicle-like structures have been observed in the host cytoplasm and have been shown to contain EMP1 [15–17]. These EMP1-containing vesicles have been reported to associate with remodeled RBC actin filaments and membranous structures, called tethers, that decorate the Maurer's clefts in the later stages of intraerythrocytic development [15,16,18].

An interesting feature of exported *Plasmodium* proteins is that they are enriched in repetitive sequences with low sequence complexity [19,20]. For example, glutamine and asparagine repeats are present in about 30% of *P. falciparum* proteins [21,22]. Proteins with low complexity sequences have been shown to play critical roles in vesicle trafficking in other eukaryotes [23,24]. Thus, it is possible that low complexity regions in exported *Plasmodium* proteins help regulate protein trafficking and host cell remodeling.

We recently reported a method for enrichment of Maurer's clefts that enabled profiling of the protein composition and identification of protein networks within this organelle [25]. That work identified several key protein clusters that play an important role in EMP1 trafficking through the Maurer's clefts. Among the clusters was an uncharacterised protein, PF3D7_0301700, that interacts with the Maurer's cleft resident protein, *Pf*EMP1 trafficking protein 6 (PTP6). PTP6 has been previously implicated in EMP1 trafficking and was shown to locate at the periphery of the Maurer's clefts [25,26].

Here we provide a detailed characterization of PF3D7_0301700, showing that it associates with the Maurer's clefts as well as chaperone-containing structures known as J-dots, and EMP1-containing electron dense vesicles (EDVs), by light and electron microscopy and immuno-precipitation. Upon deletion of PF3D7_0301700, vesicles accumulate at the Maurer's clefts; the knobs become larger and fewer in number; EMP1 delivery to the RBC membrane fails; and cytoadhesion to an endothelial cell ligand is ablated. Removing the low complexity C-terminal region of PF3D7_0301700 leads to an accumulation of vesicles at the Maurer's clefts and a defect in EMP1 trafficking. Taken together, the data show that PF3D7_0301700 plays an important role in vesicle-mediated trafficking of EMP1 to the RBC membrane. We have given it the name, *P. falciparum* EMP1 trafficking protein-7 (PTP7), to reflect its role in EMP1 trafficking.

## Results

### PTP7 is exported to the host RBC cytoplasm and Maurer's clefts

The PTP7 (PF3D7_0301700) sequence contains a predicted *Plasmodium* export element (PEXEL) (KSLAE), a recessed signal sequence (SS), an acidic N-terminal region, a central

transmembrane domain and a basic C-terminal region containing 34 consecutive asparagine residues (Fig 1A). The mature protein has a predicted molecular mass of 26,954 Da.

We used selection linked integration (S1 Fig) to generate transfectants endogenously expressing PTP7 tagged with a 2xFK506-binding protein (FKBP)-GFP-2xFKBP tag (3D7-PTP7-GFP^sand) [27]. Correct integration of the tag was verified by PCR (S1 Fig). Immunoblotting, using anti-FKBP antibodies, revealed a band with an apparent molecular mass of ~130 kDa (S1 Fig). This molecular mass is larger than the predicted 108 kDa, which may be due to the large number of basic residues in PTP7, or alternatively, due to non-skipping of the 2TA peptide [27,28,29]. A trypsin cleavage assay confirmed that EMP1 is still trafficked to the RBC surface in this cell line, albeit with fainter full length and trypsinized EMP1 fragments, possibly due to a switch in EMP1 variant expressed (S1 Fig). Likewise, electron micrographs of the parasite infected RBC confirmed that the addition of the GFP^sand tag had no effect on the Maurer's cleft ultrastructure (S1 Fig).

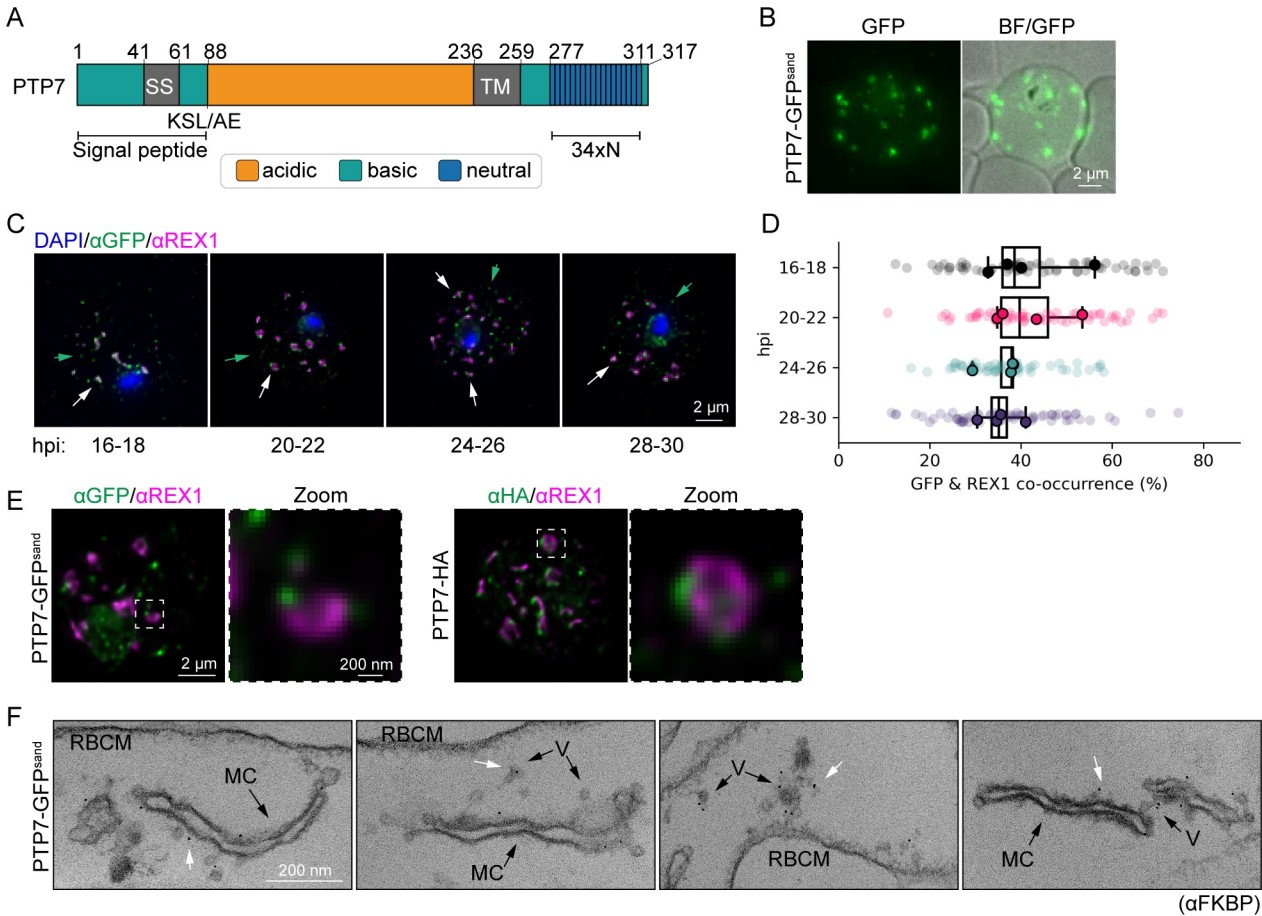

**Fig 1. PTP7 localizes to the Maurer's clefts and other compartments in the host cell.** (A) Schematic of PTP7. Amino acid numbers are indicated. SS: Signal sequence. KSL/AE: export element; TM: transmembrane domain; 34xN: 34 consecutive asparagine amino acids. (B) Live cell microscopy of PTP7-GFP^sand infected RBCs showing native fluorescence (green) merged with bright field (BF). (C) Indirect immunofluorescence microscopy of paraformaldehyde-fixed PTP7-GFP^sand infected RBCs synchronized to a 2-hour window and sampled every 4 hours from 16 to 28 hours post invasion (hpi). The infected RBCs were probed with anti-GFP (green), anti-REX1 (magenta), and DNA was stained with DAPI (blue). (D) Percentage of Maurer's clefts labeled with both anti-GFP and anti-REX1, by object segmentation. Data displayed are mean ± SD of the means per biological repeat. (E) Super-resolution microscopy of indirect immunofluorescence assays of infected RBCs synchronized to 24–26 hpi probed with the antibodies indicated. (F) Representative immuno-transmission electron microscopy (TEM) micrographs of 20–32 hpi infected RBCs permeabilized with Equinatoxin-II. Cells were probed for FKBP followed by immunogold secondary labeling. V: vesicles, MC: Maurer's cleft, RBCM: red blood cell membrane. White arrows highlight gold labelling.

Live cell microscopy of the transfectants expressing PTP7-GFP<sup>sand</sup> revealed a population of punctate structures in the host cell cytoplasm (Fig 1B). The location of PTP7-GFP<sup>sand</sup> was also assessed by immunofluorescence microscopy at four different windows from 16–30 hours post-invasion (hpi). Throughout the time-course, the PTP7-GFP<sup>sand</sup> fluorescence signal partly overlaps with that of the Maurer's cleft resident protein, ring exported protein 1 (REX1) (Figs 1C and 1D and S2), consistent with the Maurer's cleft location reported in a previous study [30].

We also tagged endogenous PTP7 with a 3xHA tag (S1 Fig) and made use of the improved resolution afforded by Airyscan super-resolution immunofluorescence microscopy. Both GFP<sup>sand</sup> and HA tagged PTP7 are revealed to associate with distinct puncta at the periphery of REX1-labeled Maurer's clefts, as well as with structures distinct from the Maurer's clefts (Figs 1E and S2). The equivalent location patterns of PTP7 tagged HA and GFP<sup>sand</sup> further provides assurance that the GFP<sup>sand</sup> tag does not alter the location of PTP7. To further investigate the sub-cellular location of PTP7-GFP<sup>sand</sup> at the ultrastructural level, trophozoite stage-infected RBCs were permeabilized with Equinatoxin II and probed with anti-FKBP antibodies. Gold particles are observed at the Maurer's clefts and at vesicle-like structures, located between the Maurer's clefts and the RBC membrane (Fig 1F, white arrows).

## PTP7 interacts with J-dot, vesicle, Maurer's cleft, and RBC membrane locating proteins

To identify proteins interacting with PTP7 we performed immunoprecipitation experiments. Mid-trophozoite stage PTP7-GFP<sup>sand</sup> transfectant-infected RBCs were solubilized in 1% Triton X-100 and associated proteins precipitated using GFP-Trap (S3 Fig). The proteins identified by mass spectrometry include PTP7 itself, the Maurer's cleft proteins, membrane-associated histidine-rich protein-1 (MAHRP1), PF3D7_0501000 and PF3D7_0601900 [31–33]; the J-dot compartment proteins, PF3D7_0801000 (0801), the gametocyte-exported protein-18 (GEXP18) and an exported HSP40 (PFE55) [34–36]; the electron dense vesicle (EDV) protein, *Pf*EMP1-trafficking protein-2 (PTP2)) [37]; and the RBC membrane skeleton protein, mature erythrocyte surface antigen (MESA). The PTP7-GFP co-precipitate was subjected to immunoblotting. Probing with antibodies to FKBP, and an antiserum raised against PTP7 (91–233 aa) (S3 Fig) confirmed pull-down of the bait. Probing with antibodies recognizing two of the interacting proteins, 0801 and MESA, confirmed their co-precipitation (S3 Fig). A network map of immunoprecipitated proteins, including data from previous studies, visualizes the protein-protein associations [25,32,38] (S3 Fig). These data suggest that PTP7 is present in multiple compartments within the RBC cytoplasm.

To confirm the presence of PTP7 in different locations in the RBC cytoplasm, dual labelled immunofluorescence microscopy was performed using antibodies recognizing GFP or HA and antibodies for antigens located in different infected RBC compartments. The co-occurrence at each location was analysed at 4 timepoints across development (S2 Fig). Image analysis reveals ~40% co-occurrence with the Maurer's cleft markers, MAHRP1 and REX1, as well as with EMP1; but also reveals ~30% co-occurrence with the Maurer's cleft/EDV-associated protein PTP2 (S2 Fig). Interestingly, the J-dot protein, 0801, shows an increasing level of co-occurrence with PTP7 as the parasite matures (S2 Fig).

## Disrupting the PTP7 locus results in aberrant knob morphology

The *ptp7* gene locus was disrupted in the CS2 parasite line using CRISPR/Cas9-directed cutting and homology-directed repair (Fig 2A). Gene deletion was validated by PCR (Fig 2B), and loss of PTP7 protein in the ΔPTP7 transfectant was confirmed by immunoblotting (Fig 2C).

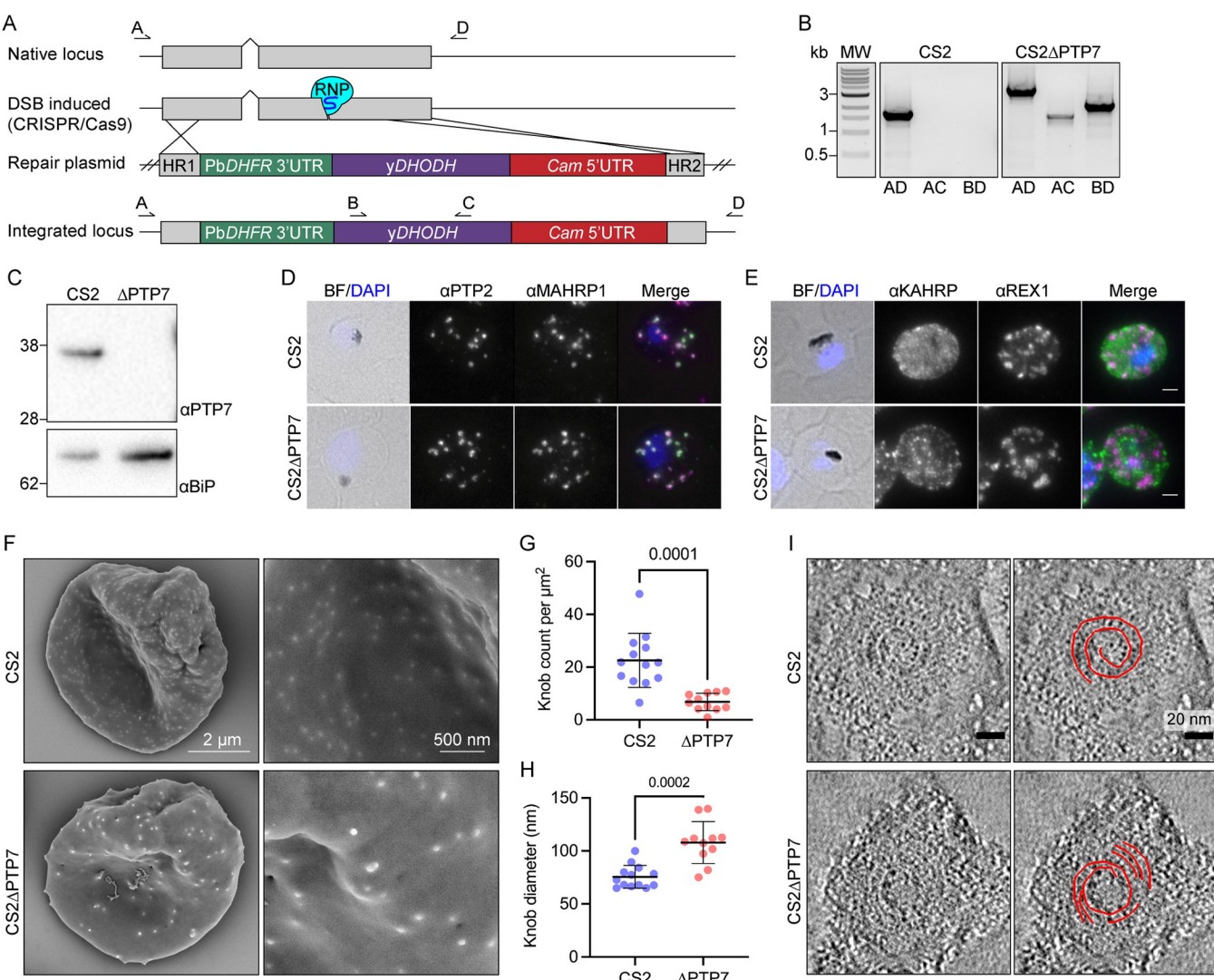

**Fig 2. Disruption of the PTP7 locus affects knob and cleft morphology.** (A) Schematic outlining the gene disruption strategy. DSB: double stranded break; RNP: ribonucleoprotein; yDHODH: yeast dihydroorotate dehydrogenase; Gray: coding sequence; up-caret: native intron; blue bulb and purple line: RNP and small guide RNA; HR1: homology region 1; HR2: homology region 2; crossing lines: homologous cross over events; arrows and letters A-D: primer locations. (B) PCR products of CS2 and CS2ΔPTP7 genomic DNA confirming disruption of the *ptp7* locus. (C) Immunoblot of cell lysates probed with αPTP7. Loading control αBiP, expected size of ~62 kDa. (D-E) Indirect immunofluorescence microscopy of acetone/methanol fixed cells. Bright field (BF) and DAPI stained DNA (blue) images are merged. Scale bar, 2 μm. (F) Scanning electron microscopy of the exterior surface of mid-trophozoite stage infected RBCs. (G) Knob density as knob count per square μm, averaged per image. Data displayed are mean ± SD (*n* = 13, 10). (H) Data points represent the average knob diameter along the major axis for knobs in an individual cell. Data displayed are the mean ± SD (CS2 *n* = 13; ΔPTP7 *n* = 10). (I) Electron tomograms of infected RBCs membranes reveal the spiral structure underlying the knob. P-values determined by Welch's t-test, *n* values are individual cells from ≥ 2 biological repeats.

Immunofluorescence microscopy of ΔPTP7 trophozoite-infected RBCs revealed no obvious change in the profiles of the Maurer's cleft markers REX1 and MAHRP1; the EDV marker PTP2 (Fig 2D and 2E); or the RBC membrane skeleton associated proteins PfEMP3, MESA and RESA (S4 Fig). In contrast, the knob protein, knob associated histidine rich protein (KAHRP), exhibited a more punctate profile when compared to wild type CS2, suggesting a disruption in knob distribution (Fig 2E). The same phenotype was observed in an independently generated CS2ΔPTP7 cell line (S4 Fig). To investigate this further, we performed scanning electron microscopy of the external surface of infected RBCs and the cytoplasmic face of

infected RBC membranes. Characteristic knob structures (76 ± 11 nm diameter) are observed on the wildtype CS2-infected RBCs, however ΔPTP7-infected RBCs exhibited fewer, larger knobs (108 ± 20 nm diameter) (Fig 2F–2H). When imaged from the cytoplasmic surface of CS2-infected RBCs, wildtype knobs appear as dimpled discs, with an average diameter of 80 ± 14 nm (S5 Fig), as described previously [3]. By contrast, the ΔPTP7-infected RBC knobs have an average diameter of 118 ± 24 nm (S5 Fig).

We employed electron tomography to investigate the organization of the spiral structure that underlies the knobs [39]. The spiral is evident, but fragmented, in ΔPTP7-infected RBCs (Figs 2I and S6), compared to more complete spirals in the CS2 samples. Taken together, these data show that both the distribution and the structure of knobs are severely compromised in the absence of PTP7. Thin section transmission electron microscopy (TEM) confirmed the deformed nature of the knob structures in ΔPTP7 parasites (S7 Fig).

### Forward trafficking of EMP1 is PTP7-dependent

We examined whether genetic disruption of PTP7 affects EMP1 trafficking and surface expression. We assessed the ability of infected RBCs to adhere to the syncytiotrophoblast ligand, chondroitin sulphate-A (CSA), under physiologically relevant flow conditions. The parent CS2 line shows efficient binding to CSA (54 ± 7 infected RBCs/field of view), however, binding to CSA is completely ablated in the ΔPTP7-infected RBCs (Fig 3A). To confirm the loss of surface displayed EMP1, we examined the ability of an antibody recognizing the ectodomain of var2CSA to label EMP1 in intact infected RBCs. A complete loss of surface labeling was observed in ΔPTP7-infected RBCs (Figs 3B and S4). Lastly, we assessed the surface accessibility of the EMP1 ectodomain to trypsin cleavage. After trypsinization of intact infected RBCs, characteristic cleavage fragments are observed, confirming the presence of surface displayed EMP1 (Figs 3C and S4). In contrast, no cleavage products are observed in the ΔPTP7-infected RBCs (Figs 3C and S4). The presence of full-length skeleton binding protein 1 (SBP1), a Maurer's cleft resident, confirms that the RBC membrane was not breached during trypsin treatment (Figs 3C and S4).

To determine where in the export pathway EMP1 was being trapped, we performed immunofluorescence microscopy using an antiserum recognizing the conserved ATS region of EMP1. This revealed Maurer's cleft labelling in both the CS2 and ΔPTP7-infected RBCs, suggesting that EMP1 can still be trafficked to the Maurer's clefts (Fig 3D, left panels). To determine if the amounts of EMP1 being trafficked to the Maurer's clefts had changed we performed a quantitative analysis of the images. We looked at the co-occurrence of EMP1 and the Maurer's clefts marker, REX1. A significant increase in the number of EMP1-containing clefts (Fig 3E, white arrows; S8 Fig) and a total increase in the amount of EMP1 at the clefts was observed in the knockout compared to the CS2 parasites (S8 Fig). Additionally, this analysis revealed that the ΔPTP7-infected RBCs exhibited fewer and larger clefts than CS2 controls (Figs 3G and S8).

### Vesicles accumulate at the Maurer's clefts in PTP7-disrupted parasites

Ultrastructural analysis of the Maurer's clefts was performed in infected RBCs that had been permeabilized with Equinatoxin II to allow introduction of antibodies [16]. In wildtype parasites, the Maurer's clefts are observed as single slender cisternae with an electron-lucent lumen and an electron-dense coat. Upon immunolabelling with anti-REX1, gold particles are observed at the cleft periphery (Fig 4A top, zoom, white arrows). Occasional protrusions are observed that may represent material budding from the Maurer's cleft surface. In ΔPTP7-infected RBCs, Maurer's clefts are still present as indicated by the gold-labelled anti-REX1 (Fig

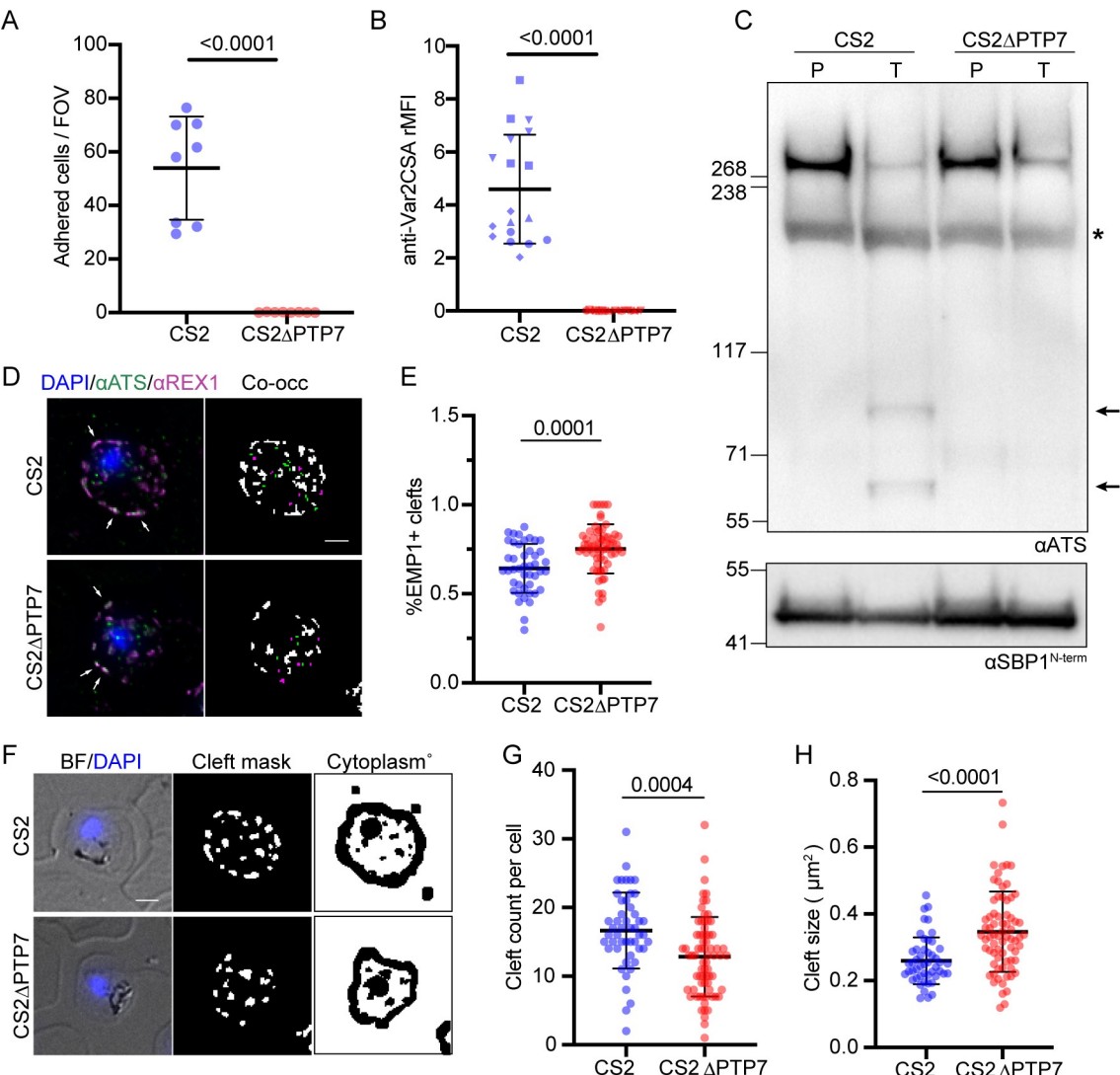

**Fig 3. PTP7 disruption affects EMP1 distribution and surface presentation.** (A) Infected RBCs were analyzed for their adherence to chondroitin sulfate A (CSA). Infected CS2 and CS2ΔPTP7 cell lines were passed through a channel slide coated with CSA under physiological flow conditions. The number of infected RBCs in 10 fields of view were recorded. Samples were run in technical triplicate and the experiment was repeated 3 times. Data displayed are mean ± SD of each technical repeat (CS2 *n* = 3; ΔPTP7 *n* = 3). (B) Flow cytometry analysis of infected RBCs labeled with antibodies recognizing the ectodomain of var2CSA, followed by secondary antibodies and tertiary antibodies conjugated to Alexa Fluor 488. Samples were run in triplicate and the experiment was repeated 3 times. The relative mean fluorescence of events was calculated as the (FITC geometric mean × number of events × $10^5$) and averaged per cell line for each biological repeat. Data displayed are mean ± SD of each technical repeat (CS2 *n* = 3; ΔPTP7 *n* = 3, symbols indicate different biological repeats). (C) Trypsin cleavage assay. Membranes are probed with αATS and αSBP1 as a control. P: PBS mock treatment; T: Trypsin treated samples; asterisk: spectrin cross-reactivity band; arrows: trypsin cleaved EMP1 products. Loading control and experimental control, αSBP1, expected molecular weight of ~50 kDa [28]. (D) Indirect immunofluorescence assays of cells fixed in acetone/methanol then probed with antibodies to the acidic terminal segment (ATS, green) and the Maurer's cleft protein REX1 (magenta) and stained for DNA using DAPI (blue). Example merged (left) and mask (right) images are shown, where EMP1 only (green), REX1 only (magenta) and both (white) objects are distinguished. White arrows indicate structures containing both REX1 and EMP1 (ATS). Scale bar, 2 μm. (E) Quantitation of the percent of EMP1 positive REX1 labeled structures. Data displayed are mean ± SD (CS2 *n* = 44; ΔPTP7 *n* = 63). (F) A representative example of the cleft and RBC cytoplasm masks used to quantify fluorescence intensity. The same cell as shown in D is represented here. (G, H) Quantification of the number and size of the Maurer's clefts in the CS2 and ΔPTP7 infected RBCs. Data displayed are mean ± SD (CS2 *n* = 41; ΔPTP7 *n* = 68). P-values determined by Welch's t-test, *n* values are individual cells from ≥ 2 biological repeats.

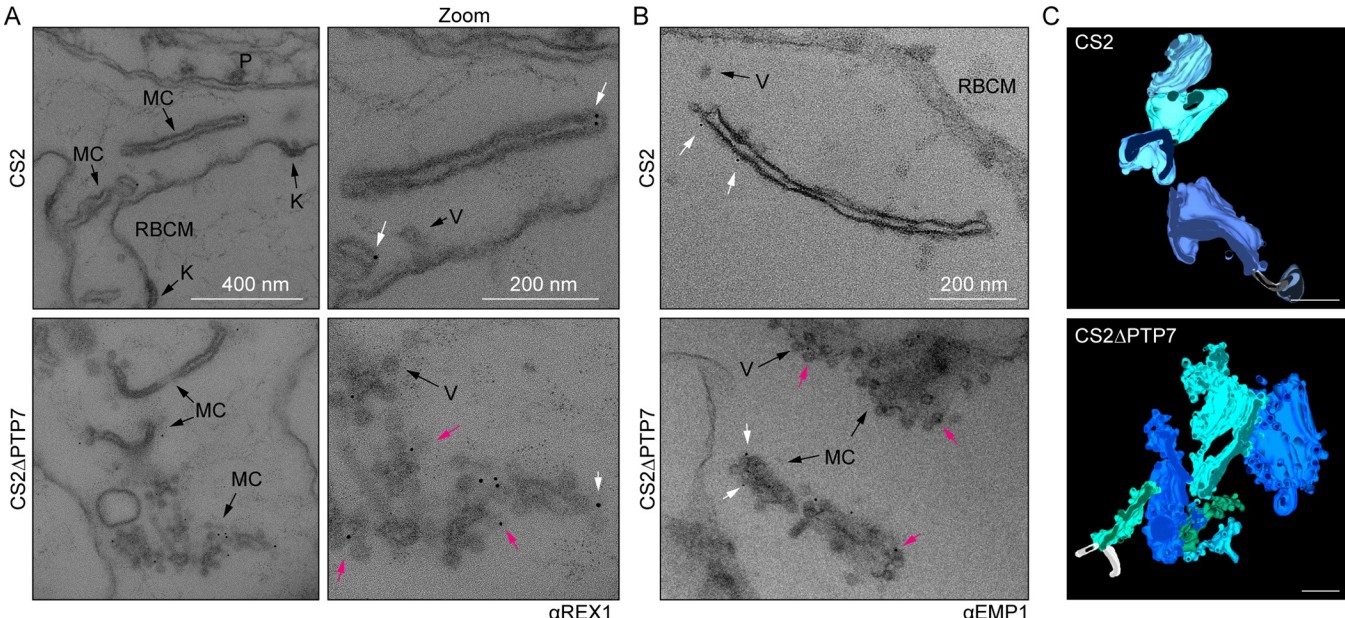

**Fig 4. PTP7 disruption leads to an accumulation of vesicles at the clefts.** (A-B) Immuno-electron micrographs of 20–32 hpi CS2 and ΔPTP7 infected RBCs permeabilized with Equinatoxin-II and probed for REX1 (A) (αREX1-repeats [63]) or EMP1 (B) (αR3031 [40]) followed by immunogold secondary labeling with protein-A (EM grade, 6 nm gold). K: knob, V: vesicle, MC: Maurer's cleft, RBCM: red blood cell membrane. (B) Immuno-electron micrographs of αR3031 (EMP1) labeled cells. Magenta arrows: gold labeled puncta, white arrows: gold labeled vesicles. (C) Clefts modeled from tomogram reconstructions of each cell line. Maurer's clefts, green and blue hues; tethers, white stalks. Scale bar, 200 nm. Translations through the tomograms and rotations of the rendered models in S1–S4 Movies.

4A bottom, zoom, white arrows). However, the clefts are decorated with numerous membrane-bound vesicle-like structures (Fig 4A bottom, zoom), with an average diameter of 44 ± 17 nm. The budding structures are morphologically similar to previously described vesicle-like structures [16] but are much more numerous than in wildtype parasites and remain in close proximity to the Maurer's clefts (Fig 4A–4C and S1–S4 Movies). The clouds of REX1 labeled vesicles around the clefts are consistent with our observation of fewer, larger clefts in ΔPTP7 (Fig 3G and 3H).

Immunolabelling with an antibody recognizing the ATS region of EMP1 [40] revealed gold particles on the body of the clefts in wildtype parasites (Fig 4B top, white arrows); and on both the body of the clefts and the associated vesicles in the ΔPTP7-infected RBCs (Fig 4B bottom, white and magenta arrows). These data suggest that these budding structures may be EMP1-trafficking vesicles that fail to separate from the clefts in the absence of PTP7.

## The C-terminal asparagine repeats are needed for PTP7 function

The majority of the PTP7 sequence is highly conserved across *Plasmodium* species, however in *P. falciparum* and *P. praefalciparum* there has been an insertion of up to 34 asparagines near the C-terminal end of the protein (see multiple sequence alignment in S9 Fig). To test the functional significance of the C-terminal extension, we generated transfectants endogenously expressing one of three truncated PTP7 species fused to a GFP reporter (Figs 5A and S10). The first truncation has amino acids 265–317 deleted, leaving the transmembrane domain and four amino acids as a linker. The second cell line contains an internal deletion removing the asparagine repeats (Δ278–310), leaving the C-terminal poly basic motif 'KKSKKN'. The final cell line expresses a truncation that removes the poly basic motif (Δ300–317). These cell lines were validated by immunoblot, confirming the presence of PTP7-GFP fusion proteins of expected

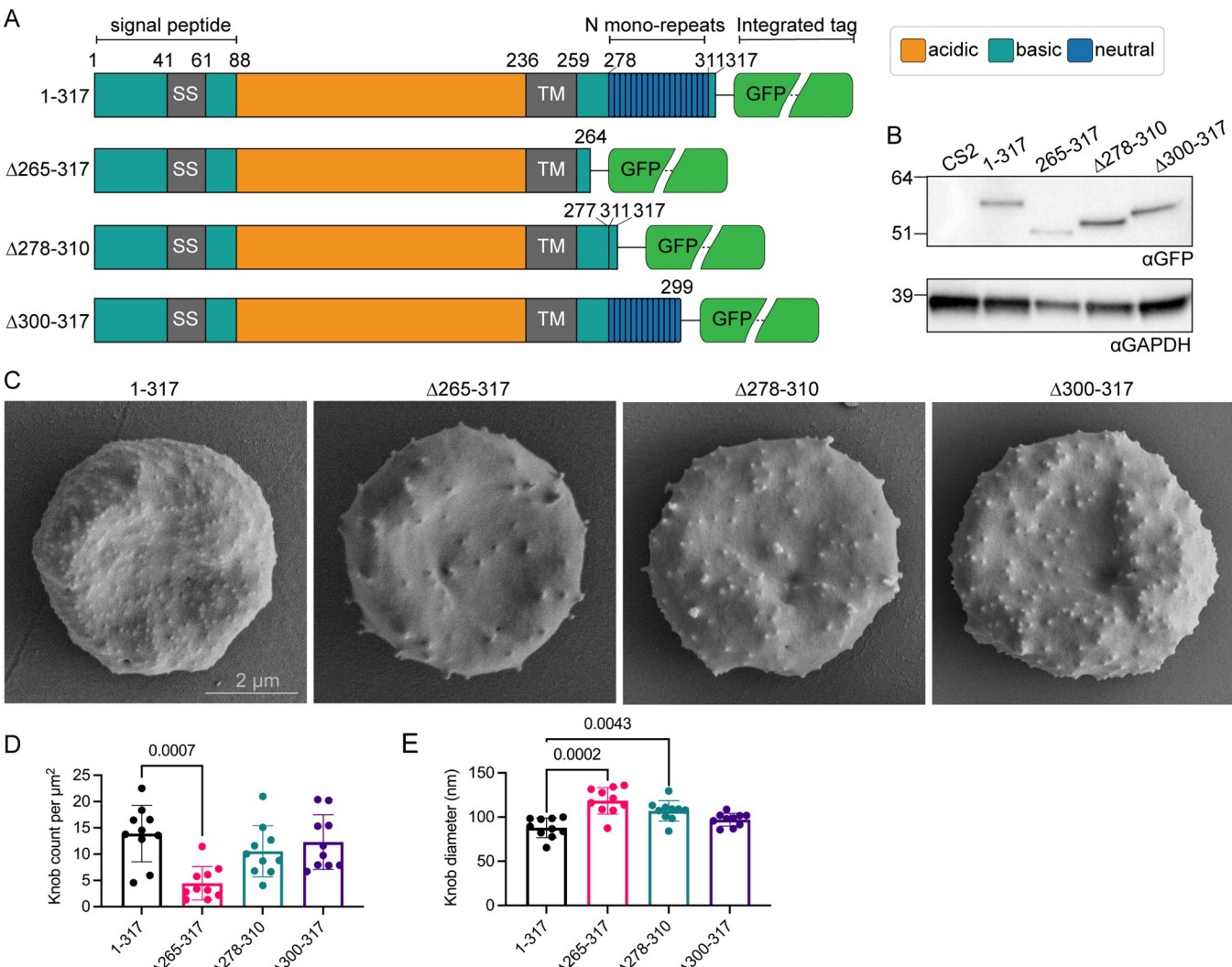

**Fig 5. PTP7 C-terminus is required for wildtype knob morphology.** (A) Schematic of the PTP7 primary sequence. Numbers indicate amino acid position. SS: Signal sequence; TM: transmembrane domain; N mono-repeats: asparagine repeats. (B) Immunoblot of lysates of the parent line (CS2) and the PTP7 C-terminally truncated lines probed with αGFP. Expected sizes of GFP chimeras from left to right are 58, 51, 54, and 56 kDa. Loading control, αGAPDH, expected size of ~38 kDa. (C) Mid-trophozoite stage infected RBCs were fixed in 2.5% glutaraldehyde/PBS and prepared for SEM of the exterior surface. Scale bar indicated. (D-E) Quantitation of the (D) knob density as knob count per square μm and (E) mean knob diameter in nm. Data displayed are mean ± SD (*n* = 10 per cell line). P-values determined by Brown-Forsythe and Welch ANOVA tests and Dunnett's multiple comparison test, *n* values are individual cells.

sizes, whereas no signal is observed in the parent CS2 cell line (Fig 5B). Detection of REX1 and KAHRP products by indirect immunofluorescence microscopy confirmed the conservation of these subtelomeric genes and PCR validation confirmed genomic integration of the constructs (S10 Fig).

Scanning electron microscopy was performed on the truncations to determine if the knob morphology was affected. Truncation of the C-terminal 17 amino acids (PTP7$^{\Delta300-317}$) had a minor effect on knob density and knob diameter (Fig 5C–5E). By contrast, deletion of the entire C-terminal region (PTP7$^{\Delta265-317}$) was associated with fewer, larger knobs (Fig 5C–5E), recapitulating the phenotype observed in the PTP7 knock-out line. Interestingly, internal deletion of the C-terminal asparagine repeats, while keeping the C-terminal 'KKSKKN' motif

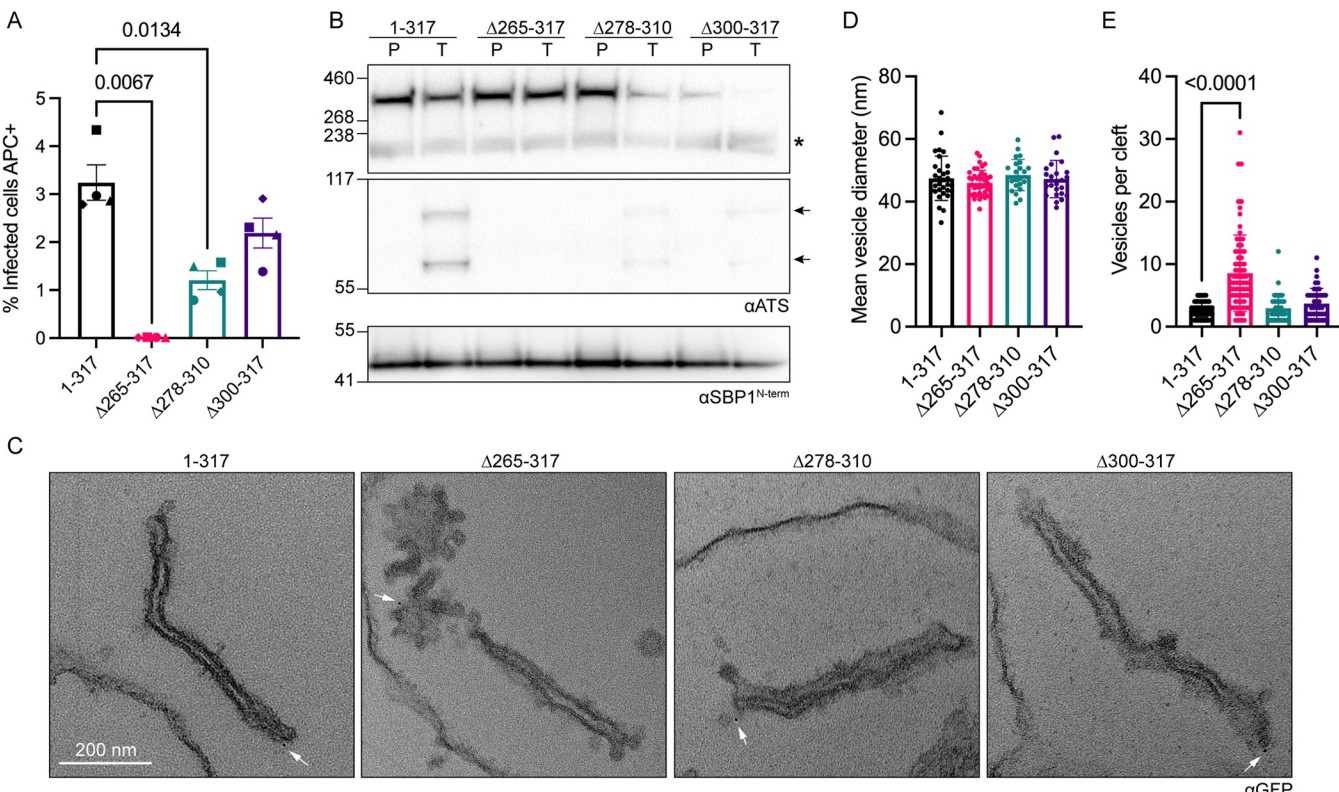

**Fig 6. A full-length C-terminus is required for antigen delivery and wildtype cleft morphology.** (A) Flow cytometry analysis of infected RBCs labeled with antibodies to the ectodomain of var2CSA followed by secondary antibodies and tertiary antibodies conjugated to Alexa Fluor 647. Samples were run in technical duplicates. Data displayed are mean fluorescence values ± SD for each biological repeat. 4 biological repeats are displayed, each with their own shape. (B) Trypsin cleavage assay of truncation lines. Membranes were probed with αATS and the loading/experimental control αSBP1. P: PBS mock treatment; T: Trypsin treated samples; asterisk: spectrin cross-reactivity band; arrows: Trypsin cleaved EMP1 products. Loading control and experimental control, αSBP1 expected molecular weight is ~50 kDa. (C) Equinatoxin-II permeabilized infected RBCs probed with αGFP followed by immunogold secondary labeling. (D-E) Quantitation of vesicles from immuno-electron micrographs. (D) Mean vesicle diameter per image. Data displayed are mean ± SD ($n$ = 32, 41, 24, 24 respectively). (E) Number of vesicles within 100 nm of each cleft. Data displayed are mean ± SD ($n$ = 43, 73, 38, 42 respectively). P-values determined by Brown-Forsythe and Welch ANOVA tests and Dunnett's multiple comparison test, $n$ values are individual cells.

(PTP7$^{\Delta 278-310}$), causes an intermediate phenotype. The PTP7$^{\Delta 278-310}$ parasites exhibit sparser knobs with larger diameters (Fig 5C–5E).

We next investigated if delivery of EMP1 to the RBC surface was affected in the truncations. Flow cytometry analysis revealed complete ablation of EMP1 at the surface of the PTP7$^{\Delta 265-317}$ infected RBCs (Fig 6A). The PTP7$^{\Delta 278-310}$ infected RBCs had a 63% reduction in surface EMP1 while the PTP7$^{\Delta 300-317}$ infected RBCs had a 32% reduction (Fig 6A). A trypsin cleavage assay was performed to confirm these results. Complete loss of EMP1 from the surface is observed in the PTP7$^{\Delta 265-317}$ infected RBCs, while faint trypsinized EMP1 bands are observed in both PTP7$^{\Delta 278-310}$ and PTP7$^{\Delta 300-317}$ parasites, consistent with the reduction of surface-exposed EMP1 detected in the flow cytometry experiments (Fig 6B). In a similar trend, the number of GFP-positive puncta is significantly reduced in PTP7$^{\Delta 265-317}$ and PTP7$^{\Delta 278-310}$ parasites (S10 Fig). These puncta counts suggest that the length of the C-terminal region of PTP7 affects its distribution.

To determine the effect that truncation of regions in the C-terminal domain of PTP7 has on vesicle accumulation at the clefts, thin section TEM was performed. Truncation of the entire C-terminal region leads to an increase in vesicles at the Maurer's clefts as revealed in the ultrastructural analysis (Fig 6C). The Maurer's clefts appear less affected when at least some of

the PTP7 C-terminal domain is retained (PTP7$^{\Delta 278-310}$ and PTP7$^{\Delta 300-317}$; Fig 6C). The vesicle diameters are not significantly affected by PTP7 mutations (Fig 6D); however, significantly more vesicles are observed within 100 nm of the PTP7$^{\Delta 265-317}$ clefts, while the PTP7$^{\Delta 278-310}$ and PTP7$^{\Delta 300-317}$ parasites exhibited a non-significant increase, relative to the full-length control (Fig 6E). The accumulation of vesicles at the clefts is similar in the PTP7$^{\Delta 265-317}$ and ΔPTP7 lines.

## Discussion

The canonical protein trafficking system that is used by most eukaryotic cells to transport proteins to the plasma membrane is not present in mature human RBCs [41]. Thus, intraerythrocytic *P. falciparum* faces the challenge of exporting integral membrane proteins, such as the virulence antigen, EMP1, to the RBC membrane; and inserting proteins into the lipid bilayer, in the absence of host cell machinery. To meet this challenge, the parasite establishes organelles called Maurer's clefts that are thought to function as intermediate trafficking compartments in the transport of EMP1 to the surface [13,14]. Moreover, several studies have described vesicle-like structures in the infected RBC cytoplasm [13,16,17,37,38,42,43]. However, the finding that EMP1 trafficking is GTP-independent [44] suggests that the molecular machinery that the parasite employs to traffic virulence proteins is divergent from that found in other cell types.

Here we characterized PTP7, a protein that was previously reported as a Maurer's cleft resident and prospective J-dot protein [30,45]. Our analysis of the location of PTP7 relative to a range of other exported *P. falciparum* proteins suggests that PTP7 localizes to different compartments in the RBC cytoplasm. Immunofluorescence labelling shows partial co-occurrence of PTP7 with the Maurer's cleft protein, REX1, throughout the intraerythrocytic cycle; with super-resolution imaging revealing that PTP7 is located at distinct puncta at the periphery of the Maurer's clefts. PTP7 also partly co-occurs with the EDV marker, PTP2, and the J-dot marker, 0801. Interestingly, an increase in the co-occurrence of PTP7 and 0801, is observed as the parasite matures.

Immunoprecipitation experiments confirm the association of PTP7 with the Maurer's cleft proteins, MAHRP1, PF3D7_0501000 and PF3D7_0601900 [31–33]; the J-dot proteins, 0801, GEXP18, and PFE55 [34–36]; the EDV marker, PTP2 [37]; and the RBC membrane skeleton associated protein, MESA. Of note, MAHRP1, EDVs and J-dots have all been implicated in EMP1 trafficking [26,37,46,47]. PTP7 localization and co-occurrence with the co-immunoprecipitants was validated using an epitope tag (PTP7-HA) and is confirmed in a recently published independent study [45].

To assess the function PTP7 plays in virulence protein trafficking, we generated two independent PTP7 knockout cell lines. Analysis of these cell lines showed that PTP7 is not essential for growth of parasites in culture, in agreement with a previous report [48]. However, in the absence of PTP7, EMP1 accumulates at the Maurer's clefts and is not presented at the surface of the infected RBC. Interestingly, upon deletion of PTP7, the Maurer's clefts became decorated with numerous vesicle-like structures that we speculate are in the process of budding from the cleft surface. Quantitative imaging revealed an accumulation of EMP1 at the Maurer's clefts in the ΔPTP7 transfectant-infected RBCs. Immuno-EM, using an antibody to EMP1, reveals labeling of both vesicles and the Maurer's clefts. Taken together, this suggests that the increase in vesicles at the clefts of the knockout infected RBCs may be linked to the defect in forward trafficking of EMP1 to the RBC membrane.

In addition to accumulation of vesicles at the clefts, the PTP7 knockout parasites also exhibit altered knob morphology, with fewer, larger knobs being assembled at the RBC surface. The main structural component of the knob complex, KAHRP, has been shown to traffic

directly to the RBC membrane, without passage through the Maurer's clefts [46]. Our data for the PTP7 knockout suggests convergence of the KAHRP and EMP1 trafficking pathways during the final step, where EMP1 is loaded into the knobs at the RBC membrane. One possibility is that the vesicles also transport a component needed for correct assembly of knobs.

It is interesting to consider how PTP7, which has no ATP or GTP binding domain, could provide the driving force for vesicle fission. In this context, the unusual asparagine repeat sequence in the C-terminal region of PTP7 may be of functional significance. While *P. falciparum* asparagine repeat sequences are dispensable in some cases [49,50], recent studies suggest that membrane-associated proteins with low complexity regions can induce membrane curvature (reviewed in [24]). In a planar membrane, disordered regions of protein are constrained, thereby reducing the conformational entropy of the protein. As a membrane curves, this constraint is relaxed and the gain in conformational entropy enables curvature sensing [51], which can support vesicle budding [52,53].

To examine the role of the PTP7 asparagine repeats in the molecular architecture of the infected RBC and in the EMP1 trafficking process, we generated transfectants in which PTP7 was truncated. Complete removal of the C-terminal domain recapitulated the phenotype seen in ΔPTP7; that is, numerous vesicles accumulate at the Maurer's clefts, while the knobs showed aberrant morphology and EMP1 trafficking is ablated. The Maurer's cleft phenotype was less dramatic if the C-terminal polybasic motif was removed (PTP7$^{\Delta300-317}$) or if the asparagine repeats were removed but the C-terminal polybasic motif was maintained (PTP7$^{\Delta278-310}$). In all truncation mutants, the trafficking of EMP1 to the surface was decreased, suggesting that the length of the C-terminal region is a factor in the efficiency of EMP1 trafficking.

Conservation of PTP7 in the *Laverania* subgenus may be linked to the expansion of virulence protein trafficking and/or host cell remodeling machinery in these species. An analysis of the currently available *Plasmodium* genome assemblies showed that the N-terminal domain of PTP7 is well conserved in *P. praefalicparum*, *P. reichenowi*, *P. gaboni*, *P. billcollinsi*, *P. alderi*, and *P. blackloci* orthologues, but the C-terminal asparagine repeats are found only in *P. falciparum* and *P. praefalciparum* [54,55] (S9 Fig). We cannot currently explain the evolution of the C-terminus in the light of what we know about the evolution of var genes in the *Laverania*, but it may suggest that the expansion of the PTP7 sequence preceded the relatively recent speciation of *P. falciparum* and *P. praefalicparum* [56]. One hypothesis is that the expanded C-terminus of PTP7 interacts with the ATS domain of EMP1, which is conserved in *P. falciparum* and *P. praefalicparum* [56], a suggestion that is supported by our truncation data showing that this region is required for EMP1 trafficking; however, additional experiments would be required to test whether the interaction of PTP7 with ATS is enhanced in the expanded PTP7 sequence.

Taken together, our data provide strong evidence that PTP7 and/or its interacting proteins are required for the recruitment or formation of EMP1-containing vesicles at the Maurer's clefts and their subsequent transfer to the host RBC membrane. Our data shows that deletion of the C-terminus of PTP7 coincides with an accumulation of vesicles at the Maurer's clefts; however, the molecular basis for this phenomenon remains unclear. One possibility is that the low complexity asparagine repeats increase the hydrodynamic radius of the C-terminal domain of PTP7 [51,52]. This increased hydrodynamic radius, in turn, may drive a preferential location at highly curved membrane structures, like the edges of Maurer's clefts and vesicles, possibly crowding these microenvironments and/or recruiting other proteins to induce vesicle fission. However, this needs to be formally tested and alternative explanations, such as a role of the C-terminus in facilitating protein-protein interactions that are required for vesicle budding are equally likely.

The interaction of PTP7 with exported proteins at several locations within the RBC cytoplasm suggest that the routes of exported protein trafficking may be more integrated and

interdependent than previously thought. This raises the alternative possibility that PTP7 functions indirectly, or in combination with other exported proteins, to facilitate vesiculation and EMP1 trafficking. Indeed, further characterization of the PTP7 interacting proteins identified in our co-immunoprecipitation experiments may reveal other exported proteins responsible for these phenotypes. An increased understanding of the processes for trafficking virulence proteins could lead to new therapies to tackle malaria pathogenesis.

## Materials and methods

### Ethics statement

Red blood cells and serum were acquired from the Australian Red Cross Lifeblood blood service. All blood products were anonymous and individual donors could not be identified. This work was approved with the written consent of the University of Melbourne Human Research Ethics Committee (approval number 1750526.3).

### *P. falciparum* culture

Parasites were cultured as described previously [25]. Briefly, *P. falciparum* cell lines were cultured in human $O^+$ RBCs (Australian Red Cross) with RPMI 1640 medium with GlutaMAX and HEPES (ThermoFisher), supplemented with 5% v/v human serum (Australian Red Cross), 0.25% w/v AlbuMAX II (ThermoFisher), 10 mM D-glucose (Sigma), 200 μM hypoxanthine (Sigma) and 20 μg/ml gentamicin (Sigma). Cultures were maintained at 5% haematocrit at 37°C in a low oxygen environment (1% $O_2$, 5% $CO_2$, and 94% $N_2$). Parasitemia was monitored by thin blood smears and Giemsa staining [57]. Cultures were synchronized at the ring stage by treatment with D-sorbitol (Sigma) as described previously [58]. Gelatin floatation with 70% Gelofusine (Braun) was used to maintain knob-positive populations of infected RBCs [59]. Transgenic parasites were maintained in the presence of selection reagent: 4–5 nM WR99210 (Jacobus Pharmaceuticals) for human dihydrofolate reductase (hDHFR) [60], 2 μg/mL blasticidin S (BSD) for blasticidin S resistance gene, 2 nM DSM1 for yeast dihydroorotate dehydrogenase (yDHODH), and 400 μg/ml G418 for neomycin phosphotransferase for SLI integration.

### Vector construction and generation of transgenic parasites

For endogenous 3' tagging, the 3' 710 bp of the *ptp7* was amplified using SLI-sand-PTP7_fw and SLI-sand-PTP7_rv primers (S1 Table) and directionally cloned into the NotI/AvrII restriction sites in the pSLI-sandwich plasmid (2xFKBP-GFP-2xFKBP) [27]. The PTP7-HA plasmid was provided by P. Gilson [45]. Ring-stage parasites were transfected with 50–100 μg of plasmid as previously described [61]. Briefly, precipitated plasmid was resuspended in sterile TE buffer and Cytomix [61]. 5–10% ring-stage infected RBCs were resuspended in the DNA mix and transferred to a 2 mm electroporation cuvette (BTX). Cells were electroporated at 310 V, 950 μF, and ∞ resistance (Gene Pulser Xcell Electroporation System, Bio-Rad), then washed in warm RPMI media and transferred to culturing dishes. After neomycin selection, correct integration of the plasmid was verified by PCR using the primers as described in S1 Fig and S1 Table.

  To generate the plasmid for CRISPR/Cas9 mediated disruption of *ptp7*, 5' and 3' homology regions (HR; ~500 bp) were PCR-amplified from genomic DNA using the primers PTP7-HR1_fw, PTP7-HR1_rv for the HR1 and PTP7-HR2_fw, PTP7-HR2_rv for HR2 (S1 Table). The HR1 fragment was cloned into the AvrII/NcoI sites and the HR2 into the SpeI/SacII sites of the pUF-TK plasmid [62]. The pUF-TK vector was linearized by digestion with

AvrII and used as the repair template. CRISPR/Cas9-mediated double-stranded breaks were guided by a single guide RNA and Cas9. The Cas9 target '*ggttccaacacagtcacacg*' was selected using CHOPCHOP and PTP7-sgRNA_top and PTP7-sgRNA_bottom oligonucleotides were annealed and cloned into the BsrgI site of pAIO using Infusion cloning (Takara Bio) [62]. Both linearized repair template and guide plasmids were transfected simultaneously into CS2 parasites. Gene disruption was confirmed by PCR using primers yDHODH_ScrF and yDHODH_ScrR, and primers in the native 5' and 3' UTRs of the *ptp7* locus, PTP7-KO_ScrF and PTP7-KO_ScrR (Fig 2 and S1 Table).

To generate the truncation parasite lines the DNA sequences encoding for amino acids 83–317, 83–264, 83–277, and 83–299 of PTP7 were PCR-amplified using the primers indicated (S1 Table) and directionally cloned into the NotI/MluI restriction sites in the pSLI-TGD plasmid to generate cell lines $PTP7^{1-317}$, $PTP7^{\Delta265-317}$, $PTP7^{\Delta278-310}$, and $PTP7^{\Delta300-317}$ respectively in a CS2 background [27]. Plasmids were transfected and endogenous integration of the truncated sequence and GFP tag were selected using neomycin as described above. Integration was confirmed by PCR using a primer upstream of the homology region, SLI-PTP7_ScrF, with SLI-PTP7-Ntrm_rv or SLI-GFP_ScrR (S1 Table). Integration of the PTP7-HA construct [45] in a CS2 background was confirmed using SLI-PTP7_ScrF, with SLI-PTP7-Ntrm_rv or HA_rv (S1 Table).

## Protein sample preparation, gel electrophoresis and immunoblotting

Whole parasite samples were prepared by lysis with saponin (Sigma). Cells were harvested and resuspended in 10 volumes of ice cold 0.03% saponin in PBS and incubated on ice for 10 min. Lysates were centrifuged at $500 \times g$ for 10 min at 4˚C, the pellet was retained, and the supernatant discarded. The pellet was washed 4 times in PBS containing cOmplete EDTA-free Protease Inhibitor Cocktail (Roche). Protein samples were mixed with 4X LDS buffer containing DTT (NuPAGE Thermofisher) and heated at 70˚C for 10 min prior to loading. Samples were separated by SDS-PAGE on either 4–12% Bis-Tris or 3–8% Tris-Acetate gradient gels (NuPAGE) run in MES, MOPS, or Tris-Acetate SDS running buffers (NuPAGE) following the manufacturers recommendations.

Protein gels were either dry transferred onto nitrocellulose membranes using the iBlot 2 device (Invitrogen) or wet transferred onto PVDF membrane using the Mini Trans-Blot Cell (Bio-Rad). Wet transfers were conducted overnight at 4˚C with ice cold transfer buffer (44 mM Tris, 30 mM glycine, 0.03% SDS, 20% v/v methanol) at 20–30 V. Membranes were blocked in 5% w/v skim milk in PBS with 0.5% Tween20 (mPBST) for at least 1 h at room temperature or overnight at 4˚C. 5% mPBST was used as the antibody diluent for all subsequent steps. Membranes were probed with the relevant primary antibody for either 90 min at room temperature or overnight at 4˚C. Primary antibodies used in this study were mouse αGFP (1:2000, 0.4 mg/ml Roche), rabbit αREX1-repeat (1:1000, [63]), mouse αATS (1:100, 1B/98-6H1-1 [28]), rabbit αFKBP (1:1000, Abcam, ab24373), mouse αKAHRP (1:1000, mAb 18.2 obtained from the European Malaria Reagent Repository), rat αHA (1:500, Roche 3F10 Sigma), rabbit αPTP7 (1:500), rabbit αSBP1^N-term (1:1000, [28]), rabbit αPTP2 (1:500, [26]), mouse αBiP (1:2000, [64]), rabbit αGAPDH (1:1000, [65]), rabbit αHSP70x (1:2000, [66]), rabbit α0801 (1:1000, [9]), rabbit αMESA (1:1000, [67]). Membranes were washed 3 times for at least 5 min each wash in PBST, then probed with anti-mouse, rabbit, or rat secondary antibody conjugated to HRP (Merck AP181P, AP132P, Invitrogen A10549) 1: 10,000–20,000 for 60 min at room temperature. Membranes were again washed 3 times in PBST before being incubated with Clarity ECL reagents (Bio-Rad) or SuperSignal West Femto Maximum Sensitivity Substrate and visualized (see S11 Fig for full length immunoblots).

## Immunoprecipitation and mass spectrometric analysis

Mid-trophozoite stage-infected RBCs were enriched by Percoll or gelatin floatation and washed twice in RPMI media. The cell pellet was solubilized on ice for 30 min with 10 volumes of immunoprecipitation buffer (1% Triton X-100 in 50 mM Tris-HCl, 150 mM NaCl, 2 mM EDTA and cOmplete EDTA-free Protease Inhibitor Cocktail (Roche)). The sample was mixed every 10 min. The samples were centrifuged at $16\,000 \times g$ for 10 min at 4°C and the supernatants were transferred to new tubes. The supernatant was again centrifuged to pellet any remaining insoluble material. The lysates were pre-incubated with Protein-A Sepharose beads (Novagen) and allowed to mix for 30 min at 4°C. The lysate was again centrifuged, and the supernatant transferred into a microcentrifuge tube containing GFP-Trap beads and mixed at 4°C for 4 h or overnight. Beads were washed 5 times in immunoprecipitation buffer and transferred to fresh tubes during the last wash. Samples were then either prepared for immunoblotting or mass spectrometry. In preparation for SDS PAGE analysis, the beads were resuspended in 4X LDS buffer containing DTT (NuPAGE Thermofisher) and warmed to 70°C for 10 min. For mass spectrometry the beads were washed twice in 1 mM Tris-HCL pH 7.4. Formic acid (0.04% final) then 2,2,2-Trifluoroethanol (7% final) were added and the samples and incubated at 50°C for 5 min. The sample was centrifuged, and the supernatant transferred to a fresh tube. Tetraethylammonium bromide (50 μM final) and Tris(2-carboxyethyl)phosphine hydrochloride (50 μM final) were added to the sample and incubated for 10 min at 70°C. Trypsin (0.25 μg) was added and the samples were incubated overnight at 37°C. Samples were subjected to tandem mass spectrometry with electrospray ionization (ESI) LC-MS/MS on a Q-Exactive mass spectrometer and analyzed as previously described [25]. Identified proteins were included if $\geq 2$ unique peptide sequences were detected and were either absent or 3-fold enriched over the control.

## EMP1 trypsin cleavage assay

Cultures containing at least 5% mid-trophozoite stage-infected RBCs were washed in PBS and divided into 2 equal fractions. Cells were harvested at $300 \times g$ for 5 min and resuspended in 10 volumes of warm PBS (P) or PBS containing 1 mg/ml TPCK-Treated Trypsin (Sigma) (T). Samples were incubated at 37°C for 1 h, inverting occasionally to prevent sedimentation. Trypsin inhibitor from *Glycine max* (soybean) (Sigma) was added to each sample to 5 mg/ml final concentration and incubated at room temperature for 15 min. Cells were harvested and the supernatant discarded. The pellet fraction was solubilized in 10 volumes of ice cold 1% TritonX-100 (Sigma) in PBS containing cOmplete EDTA-free Protease Inhibitor Cocktail (Roche) and incubated on ice for 30 min. All subsequent steps contained 1x cOmplete EDTA-free Protease Inhibitor Cocktail (Roche). Samples were centrifuged at $16\,000 \times g$ for 10 min at 4°C, then the pellets were washed three times in the ice cold TritonX-100 solution. The pellet was solubilized in 20 volumes of 2% SDS/PBS and mixed at room temperature for 30 min. The samples were centrifuged at $16\,000 \times g$ for 10 min, the supernatant transferred to a new tube and centrifuged again. The resulting supernatant was transferred to a new tube and prepared for immunoblotting.

## Antibody generation

Polyclonal antibodies against PTP7 were generated by Genscript. Briefly, the mature N-terminal domain of PTP7 (89–317 aa) was expressed in *E. coli* as a His tagged recombinant protein, which was used to immunize rabbits. Total IgG was purified from the rabbit serum using protein A.

## Live cell and immunofluorescence microscopy

For live-cell fluorescence microscopy, ~2 μl of *P. falciparum* culture was dispensed on a slide and flattened with a coverslip for immediate imaging. For indirect immunofluorescence assays, cells were either fixed in ice-cold acetone/methanol or formaldehyde/glutaraldehyde. Thin blood smears were fixed with ice-cold acetone/methanol at a ratio of either 1:1 or 9:1 for 10 min and allowed to air dry. For formaldehyde-glutaraldehyde fixation, coverslips were coated with 0.05 mg/ml *Phaseolus vulgaris* erythroagglutinin (PHA-E) lectin (Sigma) and incubated in a humid environment for 15 min at 37˚C. All subsequent incubations were conducted in a humid environment. Infected RBCs were washed in PBS, resuspended at 2% hematocrit and overlaid on the PHA-E coated slide. Cells were allowed to adhere for 15 min, then washed with PBS until a monolayer of cells was obtained. The slides were fixed in 4% formaldehyde 0.0065% glutaraldehyde in PBS for 10–20 min. The slides were washed 3 times in PBS and permeabilized with 0.1% (v/v) TritonX-100 for 10 min. The slides were washed prior to the addition of the primary antibodies diluted in PBS containing 3% (w/v) bovine serum albumin. The antibodies were incubated on the slide for 1 h at room temperature. Primary antibodies used in this study include mouse αGFP (1:1000, 0.4 mg/ml, Roche), rabbit αGFP (1:500, [68]), rabbit αREX1-rpt (1:2000, [63]), mouse αATS (1:100, 1B/98-6H1-1 [28]), mouse αEMP3 (1:500, [69]), mouse αKAHRP (1:500, mAb 18.2 obtained from the European Malaria Reagent Repository), rabbit αBand 2.1 (1:500), rabbit αPTP7 (1:500), mouse αMAHRP1c (1:200, [70]), rabbit αMAHRP2 (1:500, [18]), rabbit αPTP2 (1:200, [26]), mouse αRESA (1:500), rabbit αMESA (1:200, [67]), mouse αHA-biotin (1:500, Invitrogen 26183-BTIN). Following incubation with the primary antibody the slides were washed 3 times in PBS and the secondary antibodies added. Secondary anti-mouse or anti-rabbit antibodies conjugated to Alexa Fluor 488, 568, or 647, or streptavidin conjugated Alexa Fluor 488 (Invitrogen) (1:400 in 3% (w/v) BSA in PBS) were added to the wells for 1 h at room temperature and then the wells were washed 3 times with PBS. Each well was incubated with 2 μg/ml 4',6-diamidino-2-phenylindole (DAPI) for 15 min, then washed and mounted with p-phenylenediamine (PPD) antifade in 90% glycerol prior to sealing. Samples were imaged on a DeltaVision Elite restorative widefield deconvolution imaging system (GE Healthcare) as detailed in [25]. Briefly, samples were excited with solid state illumination (Insight SSI; Lumencor). The following filter sets (excitation [Ex] and emission [Em] wavelengths) were used: DAPI (Ex390/18 nm, Em435/48 nm), fluorescein isothiocyanate (FITC) (Ex475/28, Em523/26 nm), tetramethyl rhodamine isocyanate (TRITC) (Ex542/27, Em594/45 nm), and Cy5 (Ex632/22, Em676/34 nm). A 100X UPLS Apo (Olympus, 1.4 NA) oil immersion lens objective was used for imaging. Super-resolution confocal microscopy was performed on the LSM880 with Airyscan (Zeiss). Samples were imaged using a 63x Plan-Apochromat (Zeiss, 1.4 NA) oil immersion lens. The following Elyra laser emission and band pass filter sets were used: GFP (Ex488, Em bandpass 495–550 + lowpass 570 nm), RFP (Ex561, Em bandpass 420–480 + bandpass 495–620 nm), Cy5 (Ex642, Em bandpass 570–620 + lowpass 645 nm). Images were processed using ImageJ software [71] and deconvolved with Huygens image analysis software (Professional version 19.04, Scientific Volume Imaging) using the CMLE algorithm with 50–100 iterations and 12–15 signal to noise ratio thresholds. Image segmentation and quantification was performed using custom macros which are available from the open-access Zenodo repository (https://zenodo.org/record/6747921). Where object segmentation was required, image/object validation was performed blinded.

## Scanning electron microscopy

Whole infected RBCs were fixed and imaged as previously described [25]. Scanning electron microscopy (SEM) imaging of the cytoplasmic surface of the host RBC membranes was

performed on sheared infected RBC membranes. The membranes were prepared as previously described [25]. Briefly, glass coverslips were cleaned with acetone and 50% methanol and functionalized with 3-aminopropyl triethoxysilane (APTES), bis-sulfosuccimidyl suberate, and erythroagglutinating phytohemagglutinin (PHAE). Mature infected-RBCs were immobilized on the functionalized glass slides and sheared by forceful application of a hypotonic buffer (5 mM $Na_2HPO_4/NaH_2PO_4$, 10 mM NaCl, pH 8). For CS2 and ΔPTP7 cells, coverslips were coated with gold at 25 mA for 40 s and 75 s using a Dynavac sputter coating instrument to thicknesses of ~0.2 nm and ~0.4 nm for the sheared and whole cells, respectively. Otherwise, whole cells were gold-coated with a Safematic CCU-010 sputter coater to a thickness of 5 nm. Images were acquired with the T1 (A+B, for CS2 and ΔPTP7 whole cells) or the Everhart-Thornley detector in 'Optiplan' mode (for whole cells and sheared membranes) of an FEI Teneo SEM using a working distance of 5 mm, a beam current level of 50 pA, and 2 kV accelerating voltage.

### Immunoelectron tomography

Infected RBCs (20–32 hpi) were magnet purified, washed in PBS, and fixed in 10 pellet volumes of 2% (v/v) paraformaldehyde (PFA)/PBS for 20 min at room temperature. Cells were washed and then permeabilized in 10 pellet volumes of PBS with 1 hemolytic unit (HU) of Equinatoxin II for 6 min then washed again. Cells were then lightly fixed in 2% PFA/PBS for 5 min, washed, and incubated for 1 h with 3% BSA/PBS. The cells were incubated in one volume of the primary antibody (1:10) or no-primary control (3% BSA/PBS) for 2 h, washed, and then incubated with one volume of the gold secondary antibody for 1 h (1:15; Aurion protein A EM-grade 6-nm-diameter gold; catalog no. JA806-111). Cells were washed in 3% (w/v) BSA/PBS and then in PBS to remove the BSA.

Cell pellets were resuspended and fixed in 2.5% glutaraldehyde at 4˚C for long term storage prior to embedding in agarose, fixed in 2% osmium tetroxide (w/v) then resin embedded, sliced, and imaged as described previously [25]. Alternatively, stored cells were post-fixed in 1% osmium tetroxide (w/v) and 1.5% potassium ferrocyanide (w/v) in 0.1 M SPB (Sorensen's Phosphate Buffer, $Na_2HPO_4/NaH_2PO_4$, pH 7.4) for 30 min at room temperature in darkness. Cells were washed with 0.1 M SPB and incubated in 1% tannic acid (w/v) in 0.1 M SPB for 20 min. Cells were washed in double-distilled $H_2O$ and dehydrated via sequential 5 min incubations in 30%, 50%, 70%, 90%, (2x) absolute ethanol, and (2x) absolute acetone for 10 min each. Samples were progressively infiltrated with Procure 812 Epon-substitute resin and polymerised at 60˚C for 48 hours. Ultrathin (70 nm) and semi-thin (300 nm) sections were generated using a Leica EM Ultracut 7 ultramicrotome (Leica, Heerbrugg, Switzerland) and sections were post-stained using 2% uranyl acetate and lead citrate respectively. Imaging and electron tomography were performed on an FEI Tecnai F30 electron microscope (FEI Company, Hillsboro, OR) at an accelerating voltage of 300 kV. The tilt series were acquired for every 2˚ in the range between -70˚ and 70˚. Virtual sections were reconstructed from the raw tilt series in IMOD using a weighted back-projection algorithm [72]. Model contours were sculpted and visualized using IMOD and ImageJ.

### Infected RBC binding assay under flow conditions

Cytoadherence of infected RBCs was performed as described previously [25]. Briefly, Ibidi μ-Slide 0.2 channel slides were incubated with 100 μl chondroitin sulfate A (100 μg/ml; Sigma) in 1% BSA/PBS overnight at 37˚C. Channel slides were blocked with 1% BSA/PBS for 1 h at room temperature prior to gentle washes with 37˚C bicarbonate-free RPMI 1640 (Invitrogen). Mature infected cultures (3% parasitemia and 1% hematocrit) were harvested and resuspended

in warm bicarbonate-free RPMI 1640. Samples were pulled through the channel at 100 μl/min using a syringe pump (Harvard Apparatus) for 10 min at 37˚C to allow cytoadherence, then washed for 10 min to remove unbound cells. Adhered cells were counted at 10 points along the axis of the channel. The microscopy was performed on a DeltaVision Elite widefield imaging system (GE Healthcare).

## VAR2CSA ectodomain labeling and analysis via flow cytometry

Mid-trophozoite stage cultures were diluted to 3–5% parasitemia and 0.4% hematocrit and loaded into a 96-well plate in duplicate per condition (primary and no-primary controls). Cells were harvested at $528 \times g$ for 90 sec and washed once in 1% BSA/PBS. Cells were incubated with 10 μl of rabbit polyclonal anti-VAR2CSA antibody for 30 min at 37˚C (1:100, R1945 [40]) or 1% BSA/PBS as a control. Following incubation, the cells were washed with 1% BSA/PBS, the supernatant was discarded, and the pellets were resuspended in 10 μl of mouse anti-rabbit IgG (Sigma RG-96; 1:100) and incubated for 30 min at 37˚C. The samples were again washed in 1% BSA/PBS before addition of goat anti-mouse antibody conjugated to a fluorophore, which was incubated for 30 min at 37˚C. For CS2/ΔPTP7 experiments, tertiary antibodies were conjugated to Alexa Fluor 488. For the GFP expressing PTP7 truncated cell lines, the antibodies were conjugated to Alexa Fluor 647. Where the tertiary antibody was conjugated to Alexa Fluor 488, cells were washed and stained with the nucleic acid stain SYTO-61 as described previously [73]. Where the tertiary antibody was conjugated to Alexa Fluor 647, cells were washed and incubated with the dsDNA stain Hoechst 33342 (1:2000) for 30 min at 37˚C. Following dsDNA Hoechst labeling, the cells were washed 2 times in 1% (w/v) BSA/PBS then once in PBS prior to analysis. Flow cytometry was performed on a BD FACSCanto II with an integrated high throughput sampler. The following filter sets were used for fluorophore detection: Alexa Fluor 647 and SYTO-61 (APC, 660/20 nm), Alexa Fluor 488 (FITC, 530/30 nm) and Hoechst 33342 (Pacific Blue, 450/50 nm). A total of 50,000 events were collected and doublet discriminated. Infected cells were gated on SYTO-61/Hoechst positivity. The mean fluorescence intensities of Alexa Fluor 488/Alexa Fluor 647 fluorescence (i.e., EMP1 labeling) in these populations was calculated. Non-specific events in the secondary-only antibody controls served as a guide for drawing of the FITC+/APC+ boundary.

## Statistical analysis

Statistical tests and visualization were performed using GraphPad Prism 9.2.0 (283) Macintosh Version by Software MacKiev 1994–2021 GraphPad Software (www.graphpad.com). For comparisons between two groups only, p-values determined by Welch's t-test. For ≥2 groups, p-values determined by Brown-Forsythe and Welch ANOVA tests or Tukey's HSD test and Dunnett's multiple comparison test to the first timepoint or full-length protein. Only statistical comparisons with p-values < 0.05 are displayed.

## Supporting information

**S1 Fig. Validation and characterization of PTP7-GFP.** (A) Schematic illustrating selection linked integration of 2xFKBP-GFP-2xFKBP tag (GFP^sand) at the 3' end of the PTP7 locus. Dark gray rectangles: linkers; T: T2A skip peptide; Neo: neomycin selectable marker; hDHFR selectable marker; *PTP7: PTP7 homology region with 5' stop codon; crossed lines: homologous cross over event; letters (A-E) and half arrows: primer locations. (B) Confirmation of correct integration of the PTP7-GFP^sand plasmid into the endogenous locus. (C) Immunoblot of parent line (3D7) and PTP7-GFP^sand cell lysates probed with αFKBP. Loading control, αBiP, expected size of ~62 kDa. (D) Immunoblots of the TritonX-100 insoluble, SDS soluble fraction

of surface trypsinized infected RBCs shown. 3D7: parent line. High molecular weight band is full length EMP1 recognized by αATS. Bands annotated with an asterisk indicate spectrin cross-reactivity expected at approximately 225 kDa for polypeptides [75] and 65 kDa for spectrin degradation products [76]. Arrows indicate cleaved EMP1 species where intracellular ATS regions were protected during trypsin incubation. Loading control, αSBP1, is also an experimental control, as breaching of the infected red blood cell membrane during trypsinization would cleave the Maurer's cleft protein SBP1. SBP1 expected molecular weight of ~50 kDa [28]. (E) Representative immuno-TEM micrographs of 20–32 hpi infected RBCs permeabilized with Equinatoxin-II. Cells were probed for FKBP followed by immunogold secondary labeling. (F) Schematic illustrating selection linked integration of HA tag at the 3' end of the PTP7 locus. (G) Confirmation of correct integration of the PTP7-GFP$^{sand}$ plasmid into the endogenous locus. (H) Immunoblot of PTP7-HA and parent line CS2 cell lysates probed with αHA, fusion protein expected to run anomalously high due to low complexity regions (Fig 2C) ~41 kDa. Asterisk indicates likely degradation product. Loading control αBiP, expected size of ~62 kDa.
(TIF)

**S2 Fig. PTP7 localization late-ring to mid-trophozoite stages.** Indirect immunofluorescence assays of paraformaldehyde fixed PTP7-GFP$^{sand}$ (A) and PTP7-HA (B) infected RBCs probed with the antibodies indicated. Cells were synchronized to a 2-hour window and measured every 4 hours from 16 to 28 hpi. Scale bar, 2 μm. For quantitation of the ratio of PTP7 positive puncta to total puncta per antibody set, images were maximum projected and the parasitophorous vacuole signal was excluded. Data displayed are mean ± SD of the means per biological repeat. P-values determined by Tukey HSD multiple comparison tests ($n = 2$–4), $n$ values are biological repeats.
(TIF)

**S3 Fig. PTP7 associates with other protein trafficking structures.** (A) Summary of proteins 3-fold enriched relative to parent-line controls identified by PTP7-GFP$^{sand}$ immunoprecipitation using GFP-Trap ($n = 2$). (B) Immunoprecipitation of PTP7-GFP$^{sand}$ parasite lysate using GFP-Trap including I, input; P, post-GFP-Trap; W, resin wash; E, eluate. Expected size for PTP7 tagged with 4xFKBP 1xGFP domains is 108 kDa. Blots probed with primary antibodies against FKBP (B), PTP7 (C), 0801 (D) and MESA (E) and detected with HRP conjugated secondary antibodies. (F) Network map of exported *P. falciparum* co-immunoprecipitants identified from PTP7-GFP bait, overlayed with network established in [25,32,38].
(TIF)

**S4 Fig. Validation of the PTP7 disrupted cell lines.** (A-B) Indirect immunofluorescence assays of cells fixed at an acetone methanol ratio of 1:1 (A) or 9:1 (B) then probed with antibodies as indicated above each panel. Bright field (BF) and DAPI stained DNA (blue) images are merged. Green and magenta are used for the merged images of the two antibodies. Scale bar, 2 μm. (C) Schematic outlining the gene disruption strategy as illustrated in Fig 2. (D) PCR products of CS2 and independent knockout CS2ΔPTP7_01 genomic DNA confirming disruption of the *ptp7* locus. Red indicates saturated pixels. (E) Indirect immunofluorescence assays of cells fixed at an acetone methanol ratio 9:1 then probed with antibodies as indicated above each panel. Bright field (BF) and DAPI stained DNA (blue) images are merged. Green and magenta are used for the merged images of the two antibodies. Scale bar, 2 μm. (F) Mid-trophozoite stage infected RBCs were fixed in 2.5% glutaraldehyde/PBS and prepared for SEM of the exterior surface. (G) Flow cytometry analysis of infected RBCs labeled with antibodies for the ectodomain of var2CSA followed by secondary antibodies and tertiary antibodies

conjugated to Alexa Fluor 647. Samples were run in technical duplicates. Data displayed are mean fluorescence values ± SD for each biological repeat (*n* = 4 per cell line). (H) Trypsin cleavage assay of truncation lines. Membranes were probed with αATS and the loading/experimental control αSBP1. P: PBS mock treatment; T: Trypsin treated samples; asterisk: spectrin cross-reactivity band; arrows: EMP1 cleavage products. Loading control and experimental control, SBP1, expected molecular weight is ~50 kDa.
(TIF)

**S5 Fig. CS2 and CS2ΔPTP7 internal scanning EM images.** Early to mid-trophozoite cultures were adhered to lectin functionalized slides and sheared off with hypotonic buffer to reveal the cytoplasmic face of the infected red blood cell footprint which was then fixed, dehydrated, coated with gold, and imaged using scanning electron microscopy. Knobs are indicated by inverted discs. Data displayed are mean ± SD (CS2 *n* = 14; ΔPTP7 *n* = 12). P-values determined by Welch's t-test, *n* values are individual cells from ≥2 biological repeats.
(TIF)

**S6 Fig. Gallery of internal knob spiral tomograms.** Polylysine functionalized grids were incubated with infected RBCs then lysed and imaged. Tomograms reveal the spiral structure underlying the infected red blood cell membrane associated with knobs [39]. Scale bar, 20 nm.
(TIF)

**S7 Fig. Transmission electron micrographs of CS2 and CS2ΔPTP7.** Mid-trophozoite stage infected red blood cells.
(TIF)

**S8 Fig. Gallery of quantitation referenced in (Fig 4).** (A) IFA channels probed with αATS and αREX1 were Huygens deconvolved to resolve cytoplasmic αATS puncta and clefts indicated by αREX1 signal. A FIJI Macro script counts the αATS only, αREX1 only, and both particles to generate: a proof, including the merge image (color assignment as in (Fig 4A)) and results image with EMP1 only (green), REX1 only (magenta) and both (white) depicted. (B) Ratio of EMP1-only particles to clefts. Data displayed are mean ± SD (CS2 *n* = 44; CS2ΔPTP7 *n* = 63). (C-H) Quantitation of αATS signal distribution and cleft features. (C) Additional examples of the cleft and cytoplasm masks used to quantify the fluorescence intensities of these compartments. Bright field (BF) and DNA stain DAPI merged for reference. The 'Cleft mask' depicts where αATS signal was classified as 'cleft'. The 'Cytoplasm' mask illustrates i) the internal object used to determine αATS cytoplasmic signal and ii) the external space where αATS signal was measured as a background mean gray value control. The merge illustrates the merged IFA channels (color assignment as in (Fig 4A)). Mean gray values in the cleft and cytoplasm compartments were background subtracted. Data analysis performed is indicated in the y-axis. Data displayed are mean ± SD ((D) CS2 *n* = 42; CS2ΔPTP7 *n* = 71. (E) CS2 *n* = 45; CS2ΔPTP7 *n* = 69. (F) CS2 *n* = 41; CS2ΔPTP7 *n* = 68. (G) CS2 *n* = 49; CS2ΔPTP7 *n* = 72. (H) CS2 *n* = 49; CS2Δ85c *n* = 72.). P-values determined by Welch's t-test, *n* values are individual cells from ≥ 2 biological repeats.
(TIF)

**S9 Fig. PTP7 is conserved to *Laverania*, with asparagine repeats unique to two clades.** A phylogenetic tree and multiple sequence alignment calculated using the NCBI Standard Protein BLAST with a full-length PF3D7_0301700 (*P. falciparum* 3D7-XP_001351094.1) amino acid query. Sequences assembled in bioprojects accession: PRJNA329100, ID: 329100 and accession: PRJEB13584, ID: 445524. A hierarchical tree and color-coded sequence alignment were visualized using the ETE toolkit. NCBI provided organism names were replaced with

taxa names and identical sequences were trimmed. Only the final 58 amino acids are displayed (the transmembrane domain to the end of the C-terminus queried). Distance scale indicated. (TIF)

**S10 Fig. Validation of PTP7 truncated cell lines.** (A) Schematic illustrating selection linked integration of GFP at the 3' end of the PTP7 locus. Dark gray rectangles: linkers; T: T2A skip peptide; Neo: neomycin selectable marker; hDHFR selectable marker; *PTP7: PTP7 homology region with 5' stop codon; crossed lines: homologous cross over event; letters (A-E) and half arrows: Primer locations. (B) Confirmation of correct integration of the PTP7-GFP^sand plasmid into the endogenous locus to generate cell lines introduced in Fig 5. (C) Indirect immunofluorescence assays of paraformaldehyde fixed infected RBCs probed with the antibodies displayed. Scale bar, 2 μm. (D) Live cell microscopy, native GFP fluorescence (green) merged with the BF (gray) image. (D-E) GFP-puncta quantification of live cell microscopy showing the mean GFP-puncta counts per cell normalized to parasite width (to control for age) for each of the truncations. Data displayed are mean ± SD ($n$ = 110, 58, 26, 47 per cell line in the order displayed). P-values determined by Brown-Forsythe and Welch ANOVA tests and Dunnett's multiple comparison test, $n$ values are individual cells from ≥2 biological repeats. (TIF)

**S11 Fig. Full length immunoblots.** Saturated pixels indicated in red. (TIF)

**S1 Movie. Tomography rendering and model of CS2.** Scale bar 200 nm. (AVI)

**S2 Movie. Tomography rendering and model of CS2ΔPTP7.** Scale bar 200 nm. (AVI)

**S3 Movie. Tomography rendering and model of CS2.** Scale bar 200 nm. (AVI)

**S4 Movie. Tomography rendering and model of CS2ΔPTP7.** Scale bar 200 nm. (AVI)

**S1 Table. Oligonucleotide appendix.** (DOCX)

## Acknowledgments

The authors thank Alan Cowman, Paul Gilson, Jude Przyborski, Michael Duffy, and Stephen Rogerson for antibodies and plasmids. We thank Emma McHugh for designing the CRISPR homology region primers, and Juan Nunez-Iglesias for napari assistance. The authors gratefully acknowledge Hyun-Jung Cho, Shane Cheung, and Hamid Soleimaninejad of the Biological Optical Microscopy Platform for their support & assistance in light microscopy and image analysis. We thank Zlatan Trifunovic and Eric Hanssen for technical assistance from the Ian Holmes Imaging Centre at Bio21, The University of Melbourne (www.microscopy.unimelb.edu.au). Mass spectrometry was performed at the University of Melbourne Mass Spectrometry and Proteomics Facility, for which we thank Nicholas Williamson and Ching-Seng Ang for their assistance.

## Author Contributions

**Conceptualization:** Olivia M. S. Carmo, Leann Tilley, Matthew W. A. Dixon.

**Data curation:** Olivia M. S. Carmo, Dezerae Cox.

**Formal analysis:** Olivia M. S. Carmo, Gerald J. Shami, Dezerae Cox, Boyin Liu.

**Funding acquisition:** Leann Tilley, Matthew W. A. Dixon.

**Investigation:** Olivia M. S. Carmo, Gerald J. Shami, Boyin Liu, Adam J. Blanch, Snigdha Tiash.

**Methodology:** Olivia M. S. Carmo, Gerald J. Shami, Boyin Liu.

**Project administration:** Leann Tilley, Matthew W. A. Dixon.

**Resources:** Dezerae Cox.

**Software:** Olivia M. S. Carmo, Dezerae Cox.

**Supervision:** Leann Tilley, Matthew W. A. Dixon.

**Validation:** Olivia M. S. Carmo.

**Visualization:** Olivia M. S. Carmo.

**Writing – original draft:** Olivia M. S. Carmo, Leann Tilley, Matthew W. A. Dixon.

**Writing – review & editing:** Olivia M. S. Carmo, Gerald J. Shami, Dezerae Cox, Boyin Liu, Adam J. Blanch, Snigdha Tiash, Leann Tilley, Matthew W. A. Dixon.

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
