## [Decision Letter · Decision Letter 0]

20 Sep 2021

Dear Dr. MWA Dixon,

Thank you very much for submitting your manuscript "Virulence determinant, PTP7, controls vesicle budding from the Maurer’s clefts, adhesin protein trafficking and host cell remodeling in Plasmodium falciparum" for consideration at PLOS Pathogens. As with all papers reviewed by the journal, your manuscript was reviewed by members of the editorial board and by several independent reviewers. 

Editors and reviewers agree that PTP7 deletion gives an exciting phenotype, providing some clues about potential functions of PTP7 related to knob formation through PfEMP1 trafficking. However, at this stage of the work,  they raised key issues about the proposed model in Fig. 6F. Further experimental evidence is required to assess a direct effect of PTP7 on knob organization, including PTP7 membrane association and immunolocalization of endogenous PTP7. 

 In light of the reviews (below this email), we would like to invite the resubmission of a significantly-revised version that takes into account the reviewers' comments.

We cannot make any decision about publication until we have seen the revised manuscript and your response to the reviewers' comments. Your revised manuscript is also likely to be sent to reviewers for further evaluation.

Sincerely,

Isabelle Coppens

Associate Editor

PLOS Pathogens

Kirk Deitsch

Section Editor

PLOS Pathogens

Kasturi Haldar

Editor-in-Chief

PLOS Pathogens

orcid.org/0000-0001-5065-158X

Michael Malim

Editor-in-Chief

PLOS Pathogens

orcid.org/0000-0002-7699-2064

Reviewer's Responses to Questions

**Part I - Summary**

Reviewer #1: Parasite adhesins displayed on the surface of infected red blood cells (iRBCs) are important drivers of P. falciparum’s virulence and lethality. In the iRBC cytosol the parasite resides within the parasitophorous vacuole. Parasite proteins destined for the iRBC surface must first be trafficked through RBC cytosol. In order to facilitate this transport, the parasite generates membranous trafficking organelles in the RBC cytosol, the most important are the Maurer’s clefts.

In this paper, the parasite exported protein PF3D7_0301700 (PTP7) is shown to co-localise with REX1 (a Maurer’s cleft marker), PTP2 (an electron dense vesicle maker) and the J-dot protein 0801. Furthermore, ultrastructural imaging revealed that PTP7 containing vesicle can be seen between the Maurer’s cleft and the iRBC surface. Genetic disruptions of PTP7 were generated. In the absence of PTP7 an accumulation of vesicles at the Maurer’s Cleft was observed by TEM with immuno-labelling. In the �PTP7 parasites PfEMP1 surface presentation is disrupted and abnormal knobs are formed. This leads to only a negligible number of knockout parasites binding to the PfEMP1 cognate receptor CSA (compared to the wild type parasites).

Truncations of PTP7 phenocopy the full protein KO to varying extents. Interestingly, the removal of a low complexity c-terminal region of PTP7 resulted in negligible PfEMP1 positive cells and significantly more vesicles per Maurer’s Cleft compared to the wild type. The c-terminus of the PTP7 is clearly important for trafficking of proteins to the cell surface. The underlying mechanism this domain plays was not explored experimentally.

Overall the manuscript is well written and the experiments performed to high standard. However, I do think that the conclusions drawn and the model proposed are not sufficiently supported by data.

Reviewer #2: The manuscript describes experiments on a P. falciparum gene product previously identified as an interactor of the Maurer’s cleft protein PTP6. The gene product, named PTP7, was analysed by construction of fusion proteins (FKB-GFP and GFP) and by deleting the endogenous gene by CRISPR/Cas9 technology. Parasite lines generated with these transgenes were analysed using a variety of biochemical, immunoelectron microscopy and imaging techniques to investigate PTP7 subcellular localisation and trafficking, morphology of Maurer’s clefts and of knob structures on the surface of infected red blood cells as well as trafficking and surface exposure of the variant protein PfEMP1 as this virulence factor travels through Maurer’s cleft and vesicles originating from these strucures.

Authors conclude that PTP7 plays a key role in PfEMP1 trafficking and knob production/distribution, specifically in the process of vesicle budding at the Maurer’s cleft surface.

The experiments are generally carefully conducted with complementary techniques. Proposed role of PTP7 is an original observation in the mechanistic description of trafficking of P. falciparum virulence determinants. The proposed model for the role of repetitive sequences in governing budding dynamics is highly speculative.

Reviewer #3: The parasite driven modification of the host RBC, especially the display of antigens on the RBC membrane, is a poorly understood process because the parasite uses exported effectors for this process. These exported effectors do not share homology with proteins outside the clade and further, the RBC does not have the machinery for vesicle or membrane protein transport that could be repurposed. The manuscript by Carmo et al reveals the important role of an exported protein, PTP7, in vesicle transport within the Plasmodium infected red blood cell. The authors use several different and complimentary approaches to show that PTP7 has an essential role in the transport of the major antigenically variant protein on the infected RBC, PfEMP1. Careful microscopy studies show that PTP7 has dynamic localization through the intraerythrocytic lifecycle and this is further supported by the interaction studies. The deletion experiments suggest that an asparagine-rich and polybasic regions in PTP7 are essential for vesicle budding as well as PfEMP1 trafficking from the Maurer’s clefts. These data are consistent with their mechanistic model, that steric hinderance drives vesicle budding.

As much as I would love to see more data supporting the author’s proposed model for PTP7 in vesicle budding from Maurer’s clefts, it would require several publications. Proving this model will need careful in vitro studies with purified proteins but that is outside the scope of this excellent work. It is rare that I don’t have much to criticize in a study, this is just exceptional work. The data are state-of-the-art, well controlled, and the conclusions are very well supported. The authors should be commended on their experimental approach as well as careful analysis.

**Part II – Major Issues: Key Experiments Required for Acceptance**

Reviewer #1: The authors try to propose a model in which PTP7, via the charged residues at the C-terminus that appear specific to some Plasmodium species, acts as a driving force for vesicle fission for trafficking of PfEMP1 to the RBC surface. While the charged residues at the c-terminus could indeed play a role in membrane biology, there is no data supporting this possibility and other explanations are not well discussed. No membrane association studies have been performed and neither has the protein, or parts thereof, been expressed recombinantly and tested in liposome assays for membrane association. Therefore, in my mind all the data is showing is that deletion of PTP7, and of the C-terminal domain, leads to vesicle accumulation. I think this is a very interesting phenotype, but we have not learned how PTP7 plays a role in this. Because PTP7 also interacts with other proteins as shown by Co-IP, vesicle accumulation could also be an indirect effect and the positive amino acids are important for protein-protein interactions. These alternative explanations are not sufficiently discussed in the paper and the statement that PTP7 is a driving force in vesicle fission is not supported.

Finally, the authors say that the additional amino acids are only present in some Plasmodium species and say this is maybe related to increased virulence. I have some issues with this statement: 1) some of the laverania species genome assemblies may not be great in the regions where PTP7 lies. It would be surprising if only Pg (the most distantly related Plasmodium species to Pf) carries the extension, and none of the other species. This should be carefully examined as the authors speculate that this extension is responsible in parts for increased virulence of Pf. It also is quite surprising that PTP7 is conserved across the Plasmodium species and the non-conserved extension plays a role in a fundamental process of vesicle fusion/ or fission. What is the role of the rest of the protein then in the other species? One could test this by deleting PTP7 in Plasmodium knowlesi, but I understand this may not be easily possible.

In conclusion I think the authors provide a solid piece of experimental data but discuss the potential function of PTP7 well beyond experimental evidence. I think they should delete the model they show as it lacks sufficient data to support it and include in the discussion alternative explanations for the PTP7 phenotype (indirect effects). Alternatively, the authors could decide to invest some more time in showing membrane association of PTP7 via the polybasic stretch to support their key conclusion about the mechanism by which PTP7 functions. For sure the speculation on the evolution of PTP7 in some but not all Plasmodium species, and claims about virulence need to be supported by quality checks on the genome data.

Reviewer #2: Two key issues need to be addressed to support the authors’ conclusions.

PTP7 localisation experiments are conducted solely by analysing PTP7 fusion proteins, particularly the FKB-GFP fusion. This protein is 4 to 5 times larger than the wild type PTP7 protein, due to the presence of the reporter moieties. An independent assessment of the sites and dynamics of PTP7 localisation is needed to confirm conclusions obtained with the fusion proteins and to clear concerns that trafficking and/or accumulation to specific compartment is affected by non PTP7 sequences. PTP7 specific antibodies have been produced for this work, used only in western blot analysis; they (or additional ones in case they don't work in parasite immunostaining) should be used to investigate the subcellular localisation and traffciking of the endogenous PTP7.

The gene ablation analysis needs to be confirmed either by generation of an independent KO line and/or by functional complementation of the existing disrupted line by expression of a wild type PTP7 protein. It is assumed that destibilization of the FKB-GFP fusion have been unsuccessful.

Reviewer #3: (No Response)

**Part III – Minor Issues: Editorial and Data Presentation Modifications**

Reviewer #1: (No Response)

Reviewer #2: It is unclear whether the constructs producing the progressively deleted versions of the PTP7-GFP fusion are targeted to the ptp7 locus, ablating endogenous PTP7 expression, or elsewhere in the genome. Molecular confirmation of the chromosomal integration of these construct is needed.

Reviewer #3: 1. Were the microscopy experiments blinded before quantification?

2. How many times was the GFP-trap pulldown and mass spectrometry performed?

3. The S3 Fig. mentions 3-fold enriched proteins. How was the 3-fold abundance determined?

PLOS authors have the option to publish the peer review history of their article (what does this mean?). If published, this will include your full peer review and any attached files.

Reviewer #1: No

Reviewer #2: No

Reviewer #3: **Yes: **Vasant Muralidharan
---

## [Decision Letter · Decision Letter 1]

27 Nov 2021

Dear Dr MWA Dixon,

Editors and two previous reviewers have scrutinized your revised version of the manuscript. A reviewer remains unsatisfactory with the revisions in the Results Section, with still uncertainties about endogenous PTP7 localization and function. We all agree that data are not convincing enough in the revision to ascertain that PTP7 plays a role in vesicle accumulation at the Maurer’s cleft. Indeed, PTP7 interacts with many proteins and as stated by that authors, it may then act indirectly in virulence factor trafficking and host RBC remodeling. However, the topic is really interesting and we would like to give the opportunity to provide more experimental evidence and consider the assays proposed by the reviewer to clarify your data and justify a model at the end.

In absence of additional results, we will not publish your work at this stage . If you accept to push further your current investigation, your revised manuscript will send to reviewers for further evaluation.

Sincerely,

Isabelle Coppens

Associate Editor

PLOS Pathogens

Kirk Deitsch

Section Editor

PLOS Pathogens

Kasturi Haldar

Editor-in-Chief

PLOS Pathogens

orcid.org/0000-0001-5065-158X

Michael Malim

Editor-in-Chief

PLOS Pathogens

orcid.org/0000-0002-7699-2064

Dear Dr MWA Dixon,

Editors and two previous reviewers have scrutinized your revised version of the manuscript. A reviewer remains unsatisfactory with the revisions in the Results Section, with still uncertainties about endogenous PTP7 localization and function. We all agree that data are not convincing enough in the revision to ascertain that PTP7 plays a role in vesicle accumulation at the Maurer’s cleft. Indeed, PTP7 interacts with many proteins and as stated by that authors, it may then act indirectly in virulence factor trafficking and host RBC remodeling. However, the topic is really interesting and we would like to give the opportunity to provide more experimental evidence and consider the assays proposed by the reviewer to clarify your data and justify a model at the end. In absence of additional results, we will not publish your work at this stage of the study.

Reviewer's Responses to Questions

**Part I - Summary**

Reviewer #1: I am happy with most of the implemented changes and congratulate the authors on a very nice study! But I still find 2 aspects of the discussion should be revisited (see below)

Reviewer #2: The revised manuscript partially addressed some of the points raised in the review of the original submission.

In the discussion, the speculative mechanistic model of the PTP7 function proposed in the original submission has been removed. The sentence that “data provide intriguing evidence suggesting that the low complexity regions of PTP7 play a key role in providing the driving force for vesicle fission” is still an overstatement, but authors subsequently mention the possibility that PTP7 acts indirectly through an alternative mechanism, possibly involving protein-protein interactions.

Revision is less satisfactorily in the Results section.

**Part II – Major Issues: Key Experiments Required for Acceptance**

Reviewer #1: (No Response)

Reviewer #2: An independent determination of PTP7 localisation in the subcellular compartments of the parasitised RBC has not been provided to confirm the one obtained with the PTP7 fusion protein. Authors do explain that the anti-PTP7 antiserum used in Western blot cross reacts with several proteins, making it unsuitable to localise the wild type protein. If raising a new antibody with a tighter specificity is not feasible, a suggestion would be to produce a PTP7 fusion with a moiety structurally different from that used in the 3D7-PTP7-GFPsand parasites to confirm that both fusions show the same localisation pattern. In absence of the desirable independent confirmation, the evidence presented in the revised manuscript that Maurer’s cleft structure is unaffected in the 3D7-PTP7-GFPsand parasites is a weak argument to propose that the fusuion protein has the same localisation as the wild type PTP7. The second argument, that PfEMP1 translocation and exposure on the RBC membrane is not affected in the 3D7-PTP7-GFPsand parasites is stated by the authors (line 88) but the supporting evidence (Fig. S1) shows qualitative and quantitative differences in the pattern of the surface exposed PfEMP1 diagnostic bands obtained in 3D7 vs 3D7-PTP7-GFPsand.

The other point, i.e. to confirm the phenotype of the ptp7 gene deletion in an independent KO clone or by complementation of the one described, has been partially met. Production of an independent ptp7 KO clone is presented, but only two phenotypes are shown (Fig. S4) of those described to support the conclusions on PTP7 function, namely the reduced and dispersed presence of knobs on the surface of infected RBCs by scanning EM and the punctuate fluorescence staining obtained with the anti-KAHRP antibodies on the RBC surface.

**Part III – Minor Issues: Editorial and Data Presentation Modifications**

Reviewer #1: 1) The discussion of the PTP7 function I find is still reads one-sided, whereby the authors appear to strongly favor a model in which the c-terminus of PTP7 is directly causing vesicle fission. Again, I don’t think there is sufficient evidence for it and the statement should be made with caution. I would feel more comfortable with a statement like that: ”We currently do not understand how the c-terminus of PTP7 facilitates vesiculation at the Maurer’s clefts. One hypothesis is that it is the c-terminus itself, through an increased hydrodynamic radius, causes vesicle budding. However, this needs to be formally tested and alternative explanations, such as a role of the C-terminus in facilitating protein-protein interactions required for vesicle budding are equally likely”.

2) The authors speculate that the C-terminal extension tracks with increased virulence of Pf and P. praefalciparum compared to other Lavarania species. But are they more virulent? I don’t think we know this. This is maybe one of the most intriguing questions of the paper and I think the authors miss a great chance to discuss their findings in the light of the evolution of var genes across the Laverania (https://www.nature.com/articles/s41564-018-0162-2). Var genes are present in other Lavarania species and it begs the question why don’t they need the extension. I think it is better to say “we cannot currently explain the evolution of the c-terminus in the light of what we know about the evolution of var genes in the Laverania, but it may suggest that…”.

Reviewer #2: The sentences “minimal effect in parasite biology” and “no changes in the ultrastructure of the infected RBCs” (lines 87 and 89) are generic and imprecise to describe the results they refer to. For instance, the latter sentence specifically refers to the ultrastructure of MCs, described to be undistinguishable in the 3D7-PTP7-GFPsand line and in the wild type parasites.

PLOS authors have the option to publish the peer review history of their article (what does this mean?). If published, this will include your full peer review and any attached files.

Reviewer #1: No

Reviewer #2: No
---

## [Editor Report · Decision Letter 2]

19 Jun 2022

Dear Dr  Matthew WA Dixon,

We are pleased to inform you that your manuscript 'Deletion of the Plasmodium falciparum exported protein PTP7 leads to Maurer’s clefts vesiculation, host cell remodeling defects and loss of surface presentation of EMP1' has been provisionally accepted for publication in PLOS Pathogens. Although viable, P. falciparum lacking PTP7 exhibits an intriguing phenotype as a premise for future studies on protein trafficking in the infected RBC cytoplasm.

Best regards,

Isabelle Coppens

Associate Editor

PLOS Pathogens

Kirk Deitsch

Section Editor

PLOS Pathogens

Kasturi Haldar

Editor-in-Chief

PLOS Pathogens

orcid.org/0000-0001-5065-158X

Michael Malim

Editor-in-Chief

PLOS Pathogens

orcid.org/0000-0002-7699-2064
---

## [Editor Report · Acceptance letter]

28 Jul 2022

Dear Dr Dixon,

We are delighted to inform you that your manuscript, "Deletion of the Plasmodium falciparum exported protein PTP7 leads to Maurer’s clefts vesiculation, host cell remodeling defects, and loss of surface presentation of EMP1," has been formally accepted for publication in PLOS Pathogens.

Best regards,

Kasturi Haldar

Editor-in-Chief

PLOS Pathogens

orcid.org/0000-0001-5065-158X

Michael Malim

Editor-in-Chief

PLOS Pathogens

orcid.org/0000-0002-7699-2064